# Heterogeneity measures in hydrological frequency analysis: review and new developments

Ana I. Requena [1], Fateh Chebana [1], Taha B. M. J. Ouarda [2,1]

[1] Institut National de la Recherche Scientifique (INRS), Centre Eau Terre Environnement (ETE), Quebec, G1K-9A9, Canada
[2] Institute Center for Water and Environment (iWATER), Masdar Institute of Science and Technology, Abu Dhabi, PO Box 54224, United Arab Emirates

*Correspondence to*: Ana I. Requena (ana.requena@ete.inrs.ca)

**Abstract:** Some regional procedures to estimate hydrological quantiles at ungauged sites, such as the index-flood method, require the delineation of homogeneous regions as a basic step for their application. The homogeneity of these delineated regions is usually tested providing a yes/no decision. However, complementary measures that are able to quantify the degree of heterogeneity of a region are needed to compare regions, evaluate the impact of particular sites and rank the performance of different delineating methods. Well-known existing heterogeneity measures are not well-defined for ranking regions, as they entail drawbacks such as assuming a given probability distribution, providing negative values and being affected by the region size. Therefore, a framework for defining and assessing desirable properties of a heterogeneity measure in the regional hydrological context is needed. In the present study, such a framework is proposed through a four-step procedure based on Monte Carlo simulations. Several heterogeneity measures, some of which commonly known, others derived from recent approaches or adapted from other fields are presented and developed to be assessed. The assumption-free Gini Index applied on the at-site L-variation coefficient (L-CV) over a region led to the best results. The measure of the percentage of sites for which the regional L-CV is outside the confidence interval of the at-site L-CV is also found to be relevant, as it leads to more stable results regardless of the regional L-CV value. An illustrative application is also presented for didactical purposes, through which the subjectivity of commonly used criteria to assess the performance of different delineation methods is underlined.

*Keywords*: hydrology; regional analysis; ungauged site estimate; heterogeneity degree; L-variation coefficient; Gini Index.

## 1 Introduction

Regional hydrological frequency analysis (RHFA) is needed to estimate extreme hydrological events when no hydrological data are available at a target site or to improve at-site estimates especially for short data records (e.g. Burn and Goel, 2000; Requena et al., 2016). This is usually done by transferring information from hydrologically similar gauged sites. Delineation of regions formed by hydrologically similar gauged sites is a basic step for the application of a number of regional procedures such as the well-known index-flood method (Dalrymple, 1960; Chebana and Ouarda, 2009). Such a method employs information from sites within a given "homogeneous" region to estimate the magnitude of extreme events related to

a given probability (or return period) at a target site, which are called quantiles. Regional homogeneity is often defined as the condition that floods at all sites in a given region have the same probability distribution except for a scale factor (e.g. Cunnane, 1988). The present paper focuses on the heterogeneity concept in hydrology derived from this 'regional homogeneity', which is different from the heterogeneity concept considered in other fields, such as ecology, geology and information sciences (e.g. Li and Reynolds, 1995; Mays et al., 2002; Wu et al., 2008).

In order to delineate homogeneous regions, numerous studies have proposed and compared similarity measures entailing climatic (e.g. mean annual rainfall), hydrologic (e.g. mean daily flow), physiographic (e.g. drainage area) and combined descriptors (see Ali et al., 2012 and references herein) to be used as input to statistical tools for grouping sites. The selection of these descriptors is carried out by stepwise regression, principal components or canonical correlation, among others (e.g. Brath et al., 2001; Ouarda et al., 2001; Ilorme and Griffis, 2013). Traditional statistical tools, such as cluster analysis, or new approaches, such as the affinity propagation algorithm, are considered to form homogeneous regions based on the previously identified similarity measures (e.g. Burn, 1989; Ouarda and Shu, 2009; Ali et al., 2012; Wazneh et al., 2015). For further references on regional flood frequency analysis, please see Ouarda (2013), Salinas et al. (2013) and references herein. Moreover, many tests have been introduced and compared throughout the literature to decide whether a given delineated region can be considered as homogenous (e.g. Dalrymple, 1960; Wiltshire, 1986; Scholz and Stephens, 1987; Chowdhury et al., 1991; Fill and Stedinger, 1995; Viglione et al., 2007). The homogeneity test proposed by Hosking and Wallis (1993) is usually utilised. In this test the statistic $H$ is related to the variability of the at-site L-variation coefficient (L-CV) over a region (e.g. Alila, 1999; Burn and Goel, 2000; Castellarin et al., 2001; Shu and Burn, 2004; Smith et al., 2015; Ouarda, 2016).

In practice, apart from determining if a region can be considered as homogeneous by making a yes/no binary decision (e.g. Warner, 2008) generally based on a significance test, the quantification of the degree of heterogeneity is also necessary. Heterogeneity measures are required for such a task. Two approaches can be considered in this regard: (i) the use of heterogeneity measures for determining the effect of the departure from the homogeneous region assumption on quantile estimation; and (ii) the use of heterogeneity measures for ranking regions according to their degree of heterogeneity. Regarding the former, quantifying the degree of heterogeneity provides a notion of the inaccuracy incurred through the estimation of quantiles by a regional method, for which homogeneous regions are assumed but a 'non-perfect' homogeneous region is used. This approach has already been studied, being closely related to the homogeneity test notion (e.g. Hosking and Wallis, 1997; Wright et al., 2014), which is further explained below.

The second approach corresponds to the focus of the present paper. Through this second approach, different regional delineation methods can be properly compared to identify the best one. This will be the method delineating the 'most homogeneous region'. Also, heterogeneity measures can be helpful in ranking potential homogeneous regions formed by removing discordant sites. By analogy with distribution selection (e.g. Laio et al., 2009), the concept of heterogeneity measure considered here plays the role of a 'model selection criterion', such as the Akaike Information Criterion (Akaike, 1973); whereas the homogeneity test plays the role of a 'goodness-of-fit test'. The former ranks delineated regions by

providing unambiguous results to identify the best one in terms of heterogeneity; whereas the latter indicates if the given region can be considered as homogeneous or not.

In relation to the use of heterogeneity measures as a proxy for quantile error (approach (i)), the test statistic $H$ is indeed considered by Hosking and Wallis (1993) as a heterogeneity measure for which given thresholds are established. These thresholds are obtained as a trade-off between quantile error due to regional heterogeneity and gain obtained by considering the whole regional information instead of that of a sub-region or at-site data. Therefore, instead of providing a binary decision based on a given significance level (α), e.g. reject the region as homogenous when $H > 1.64$ for α = 5%; as a more general guideline the region is considered as 'acceptably homogeneous' if $H < 1$, 'possibly heterogeneous' if $1 \le H < 2$ or 'definitely heterogeneous' if $H \ge 2$. Recently, Wright et al. (2015) compared the performance of five statistics in this regard: the three L-moment-based statistics of Hosking and Wallis (1993) and two non-parametric statistics, the Anderson-Darling and the Durbin-Knott test statistic.

A number of studies have proposed and compared methods in which different combinations of similarity measures and/or statistical tools are considered for delineating regions (references below). These studies usually consider measures based either on $H$ or on errors from the quantile estimation step. The reason is the non-availability of a well-justified heterogeneity measure for comparison purposes (approach (ii)). Shu and Burn (2004) utilised the percentage of (initially) homogeneous regions and the mean of $H$ over regions obtained by each considered method for distinguishing the best one. Farsadnia et al. (2014) identified the best grouping method among those analysed as that leading to the lowest number of 'possibly homogeneous' and 'heterogeneous' regions according to $H$. Ilorme and Griffis (2013) used an $H$ weighted average regarding the data length of each region to compare regions obtained by removing discordant sites based on different criteria.

However, $H$ is not well-defined for ranking regions according to their heterogeneity degree, as it possesses several drawbacks. First, it is originally built as a significance test. Thus, its value depends on specific assumptions that may not be fulfilled in practice, such as assuming a regional kappa distribution that even though flexible may not characterise the data. Second, it may entail negative values for particular situations, which may distort results making difficult the suitable ranking of regions. Third, it is affected by the number of sites in the region, tending to obtain small heterogeneity values for small regions even if they are not homogeneous (Hosking and Wallis, 1997, page 66-67). This tends to complicate comparison among regions with different sizes.

Instead of using measures based on $H$, other studies quantified the performance of different delineating methods by comparing quantile errors (e.g. Castellarin et al., 2001; Ouali et al., 2016). However, comparing quantile errors implies performing the last step of a regional analysis (i.e. quantile estimation) when dealing with an initial step (i.e. region delineation); which involves additional calculations, uncertainty due to the assumption of a given parent distribution for the data and a non direct assessment of the delineation method. A different approach was recently proposed by Viglione (2010) and Das and Cunnane (2011) regarding the use of the confidence intervals for L-CV to assess heterogeneity, for which details are given in Sect. 3.

Therefore, a general framework is needed to allow defining and assessing desirable properties of a heterogeneity measure in the regional hydrological context in order to properly identify a suitable measure. Such a measure should overcome the aforementioned drawbacks: it should be free of assumptions, positive and not affected by region size. Furthermore, the use of a heterogeneity measure should allow direct comparison of the heterogeneity of regions delineated by different methods. Indeed, it should allow ranking the heterogeneity degree of several regions to identify 'the most homogeneous region' or to assess the effect of some sites on the 'heterogeneity degree' of the region. In the present paper, such a framework is proposed under an evaluation of the heterogeneity measures based on Monte Carlo simulations. Several measures extracted from literature in hydrology and other fields are presented and/or adapted to be assessed as well-justified heterogeneity measures. The present paper is organised as follows. The procedure for the assessment of a heterogeneity measure is presented in Sect. 2. The heterogeneity measures considered to be checked by the proposed procedure are introduced in Sect. 3. Results of the assessment are illustrated in Sect. 4. Discussion of results is presented in Sect. 5. An illustrative application is shown in Sect. 6 and conclusions are summarised in Sect. 7.

## 2 Assessment of a heterogeneity measure

A simulation-based procedure consisting of four steps is proposed to study the behaviour of a given heterogeneity measure (generically denoted $Z$) regarding its desirable properties in the regional hydrological context. The steps of the procedure are: (i) sensitivity analyses of varying factors involved in the definition of a region; (ii) success rate in identifying the most heterogeneous region; (iii) evolution of the variability for the $Z$ average with respect to the degree of regional heterogeneity; and (iv) effect of discordant sites. The first step is applied to all the studied heterogeneity measures (presented in Sect. 3) and may be considered as preliminary, while the remaining steps are applied to those not entailing unacceptable results from the first step. Some elements of the procedure are inspired and adapted from studies where different aims were sought (e.g. Hosking and Wallis, 1997; Viglione et al., 2007; Chebana and Ouarda, 2007; Castellarin et al., 2008; Wright et al., 2015).

### 2.1. Synthetic regions

Before further describing the aforementioned steps and desirable properties, elements of the framework needed for performing the assessment procedure are presented. The procedure is based on synthetic regions with flood data samples generated through Monte Carlo simulations from a representative flood parent probability distribution commonly used in frequency analysis, the Generalised Extreme Value (GEV) distribution. A region is defined by its number of gauging sites ($N$), at-site data length ($n$), regional average L-CV ($\tau^R$), regional average L-skewness coefficient ($\tau_3^R$) and a unit regional sample mean. The heterogeneity of a given region may be due to differences in any feature of the at-site frequency distribution among sites. In particular the L-CV, which is a dimensionless measure of the dispersion of the distribution that is also related to the slope of the associated flood frequency curve, has been considered as representative of such differences (e.g. Stedinger and Lu, 1995; Viglione, 2010). In the present study, heterogeneous regions are simulated using the

heterogeneity rate $\gamma$, defined as $\gamma = (\max_i(\tau^i) - \min_i(\tau^i))/\tau^R$ (e.g. Hosking and Wallis, 1997; Das and Cunnane, 2012),

where $\tau^i$ is the L-CV at site $i$ with $i = 1, \ldots, N$. Since in practice large values of the L-skewness coefficient ($\tau_3$) are related to large values of the L-CV $\tau$, and based on studies in the literature (e.g. see Hosking and Wallis, 1997, page 68 and Table 4.1; Viglione et al., 2007, Figure 1), the same heterogeneity rate of $\tau$ is considered for $\tau_3$. A region is defined as homogeneous for

$\gamma = 0\%$, implying that $\tau^i$ and $\tau_3^i$ are the same for all the sites in the region (i.e. $\tau^i = \tau^R$ and $\tau_3^i = \tau_3^R$). The heterogeneity of a given region increases as $\gamma$ increases from 0% to 100%. This implies that $\tau^i$ and $\tau_3^i$ vary linearly. We then have for the first site $\tau^1 = \tau^R - \tau^R \gamma/2$ and for the last site $\tau^N = \tau^R + \tau^R \gamma/2$. The same can be written for $\tau_3^i$. Note that this relation is commonly used in other studies (e.g. Hosking and Wallis, 1997; Wright et al., 2015) as a plausible way of simulating varying conditions over a region.

Finally, a given region consists of at-site data generated from a GEV distribution with parameters obtained through at-site L-moments. At-site data are standardised by their sample mean to frame them in the regional context (e.g. Bocchiola et al., 2003; Requena et al., 2016). Note that heterogeneity measures directly based on L-moments lead to the same results for standardised or non-standardised data. A region with $N = 15$, $n = 30$, $\tau^R = 0.2$ and $\tau_3^R = 0.2$ is considered as a reference for the simulation study. Hereafter, the value of $\tau_3$ is (usually) omitted, as $\tau_3$ is considered to have the same value as $\tau$ (e.g. Hosking

and Wallis, 1997). The number of simulations $N_S$ of a given region is taken to be equal to 500, which is considered large enough to obtain robust results. These fixed values of the factors, as well as their varying values used below, are selected according to the literature and with the aim of providing a general view of the behaviour of the measures without excessively complicating the simulation study.

It is important to highlight that the use of simulated data in the assessment of new techniques in regional frequency analysis

is a well-established approach and it has been used in a number of publications (e.g. Hosking and Wallis, 1997; Seidou et al., 2006; Chebana and Ouarda, 2007).

## 2.2. Sensitivity analyses

The first step of the assessment of a heterogeneity measure $Z$ is the analysis of the effect of varying factors involved in the definition of a region. This step is performed through sensitivity analysis to identify if the behaviour of $Z$ is acceptable in

relation to what is ideally expected from a heterogeneity measure.

**Effect of the heterogeneity rate:** The degree of heterogeneity of a region is the aimed value to be quantified by $Z$. A surrogate of such a degree of regional heterogeneity is the heterogeneity rate $\gamma$, which is used to initially define the heterogeneity of the simulated region to be evaluated by $Z$. Hence, $Z$ should increase with $\gamma$. This analysis is performed by obtaining $Z$ for $\gamma = 0\%$, 10% ,…, 90%, 100% over $N_S = 500$, keeping the remaining values of the reference region (i.e. $N = $

15; $n = 30$; $\tau^R = 0.2$).

**Effect of the number of sites:** The size of a region, represented by the number of sites $N$, is a relevant factor to the degree of its heterogeneity. A large $N$ is required to properly estimate quantiles associated with high return periods, as more data are

available; yet homogeneous regions are more difficult to obtain for large $N$ due to more potential dissimilarities between sites (Ouarda et al., 2001; Chebana and Ouarda, 2007). Nevertheless, by definition $Z$ should not be affected by $N$, as it should provide the same results for regions with a different size but the same degree of heterogeneity. Therefore, the smaller is the influence of $N$ on $Z$ the better $Z$ is. This analysis is performed by obtaining $Z$ for $N = 5, 10, 15, 20, 25, 30, 40, 50, 60,$ 70 over $N_S = 500$, keeping the remaining values of the reference region (i.e. $n = 30$; $\tau^R = 0.2$). Two different values of the heterogeneity rate ($\gamma = 0\%$ and $50\%$) are also considered to identify if the behaviour of $Z$ changes depending on the degree of heterogeneity.

**Effect of the regional average L-moment ratios:** $Z$ should ideally provide similar results for regions entailing the same degree of heterogeneity, regardless of the values of $\tau^R$ and $\tau_3^R$, in order to provide an appropriate comparison and ranking of the regions. For instance, two regions with sites generated from a different $\tau^R$ value but considering the same value $\gamma = 0\%$ should entail similar $Z$ values, as both are 'perfectly' homogeneous. However, such an output may not be easy to obtain due to the fact that $\tau^R$ is associated with a measure of dispersion. Thus, the smaller the influence of $\tau^R$ and $\tau_3^R$ on $Z$ the better $Z$ will be. This analysis is performed by comparing the results of $\tau^R = 0.2$, which is related to the reference region, with those obtained by $\tau^R = 0.4$. It is done by varying the heterogeneity rate $\gamma$ and by varying the number of sites $N$. Recall that $\tau_3^R$ is considered to have the same value as $\tau^R$.

**Effect of the record length:** The amount of available at-site information, represented by the data length $n$, is associated with the accuracy of the value of $Z$. The longer $n$ is the better will be the information provided by each site to determine the regional degree of heterogeneity. Therefore, the analysis of the effect of $n$ should be focused on identifying the minimum $n$ required to obtain reliable values of $Z$. This analysis is performed by obtaining $Z$ for $n = 10, 20,\ldots, 90, 100$ over $N_S = 500$, keeping the remaining values of the reference region (i.e. $N = 15$; $\tau^R = 0.2$). Two different values of the heterogeneity rate ($\gamma = 0\%$ and $50\%$) are also considered to identify if the behaviour of $Z$ changes depending on the degree of heterogeneity.

### 2.3. Success rate

The second step in the assessment of $Z$ is the evaluation of its success rate ($SR$) for identifying the most heterogeneous region. Note that the $SR$ notion is commonly used in a number of fields such as biology (e.g. Canaves et al., 2004). Without loss of generality, such an evaluation is performed on two regions A and B. For $\gamma_A < \gamma_B$, $SR$ is defined as the ratio of the number of samples simulated from a given region A and a given region B, for which $Z$ correctly identifies $\gamma_B$ as the most heterogeneous region, to the total number of simulated samples. Thus, the larger $SR$ is the better $Z$ will be. The aim is to verify the ability of $Z$ to compare regions with different degrees of heterogeneity, when entailing or not different characteristics (i.e., $\tau^A \neq \tau^B$ or $\tau^A = \tau^B$, and $N_A \neq N_B$ or $N_A = N_B$). A large set of 48 cases is considered to obtain a wide view of the behaviour of $Z$, as combination of the following factor values: $\gamma_A = 0\%, 30\%, 50\%, 70\%$ with $\gamma_B = \gamma_A+10\%,$ $\gamma_A+20\%, \gamma_A+30\%; N_A = N_B, N_A \neq N_B$ (for $N = 10, 25$); $\tau^A = \tau^B, \tau^A \neq \tau^B$ (for $\tau^R = 0.1, 0.2, 0.3, 0.4$) over $N_S = 500$, keeping the remaining values of the reference region (i.e. $n = 30$).

## 2.4. Evolution of the variability for the $Z$ average with respect to the degree of regional heterogeneity

The third step of the assessment of $Z$ is the analysis of the evolution of the variability of the average value of $Z$ as a function of the degree of regional heterogeneity. The aim is to determine the capability of $Z$ to accurately rank regions according to their degree of heterogeneity when it is summarised as an average of the $Z$ values obtained for several (sub)regions that are obtained by a given delineation method. This provides an assessment of its capability to compare results from several delineation methods. This is a twofold analysis. Firstly, a monotonic relation should exist between the average $Z$ and the degree of heterogeneity, as explained in Sect. 2.2. Secondly, the variability of the average $Z$ along such a monotonic relation should be small enough to not affect a proper ranking of the regions.

We consider two regions A and B, without loss of generality. The idea is that (sub)regions delineated by a given method should theoretically entail different $\tau^R$ values ($\tau^A \neq \tau^B$), having similar or different values of other characteristics (i.e. $N_A \neq N_B$ or $N_A = N_B$). In order to be able to evaluate the behaviour of the $Z$ average, the same degree of heterogeneity is considered for both regions ($\gamma_A = \gamma_B = \gamma$), as under this assumption $Z$ values should be similar. The procedure is the following: $N_S = 500$ simulated regions A and B with $\gamma_A = \gamma_B = \gamma$ and given values $N_A$, $\tau^A$ and $N_B$, $\tau^B$ are generated, obtaining for each simulation the average of $Z$ over the two regions. These averages are aggregated into their mean value over $N_S$ as representative value. The representative value is obtained for 22 cases as a result of combining: $N_A = 10, 25$; $N_B = 10, 25$; and $\tau^R = 0.1, 0.2, 0.3, 0.4$ with $\tau^A \neq \tau^B$, keeping the remaining values of the reference region (i.e. $n = 30$). Then, the variability of the set of representative values of the $Z$ average is analysed through a boxplot for the given $\gamma$. The aforementioned procedure is performed for each $\gamma = 0\%, 10\%, \ldots, 90\%, 100\%$, obtaining a boxplot for each $\gamma$ value. For a given $\gamma$, $Z$ is better as the variability of the corresponding set of representative values is smaller, since similar values of $Z$ should be expected due to $\gamma_A = \gamma_B$. Then, $Z$ is better as the interquantile range is shorter, where the interquantile range is the box of the boxplot. For varying $\gamma$, $Z$ is better as it does not imply overlapping of the interquantile ranges for different $\gamma$ values, which leads to a more appropriate ranking of the regions.

## 2.5. Effect of discordant sites

The fourth step of the assessment of $Z$ is the analysis of the effect of discordant sites in a region. The aim is to check the capability of $Z$ to show a progressive variation of its value as a consequence of a progressive change in the degree of regional heterogeneity, induced here by replacing given 'homogeneous' sites by given 'discordant' sites in a region. Both the effect of the 'nature' of the discordant sites, characterised by the L-CV $\tau^d$ and L-skewness coefficient $\tau_3^d$ of their parent distribution, and the effect of the number of such discordant sites ($k$) are considered.

The procedure is described below. Note that the values of the factors used in this section are selected to facilitate the graphical representation. Thus, a homogeneous region (i.e. $\gamma = 0\%$) with $N = 20$, $\tau^R = 0.25$ and $n = 30$ is considered as the initial region. Then, $k$ of its sites (with $k = 1, \ldots, N/2$) are replaced by $k$ discordant sites belonging to a parent distribution characterised by $\tau^d$, with $\gamma_d = 0\%$ within the group of discordant sites. The analysis is performed for $\tau^d = 0.1, 0.2, 0.25, 0.3,$

0.4. Remark that $\tau^R = 0.25$ is considered for the homogeneous region so that the discordant sites are not 'discordant' at the midpoint of the range used for $\tau^d$ (i.e. at $\tau^R = \tau^d = 0.25$). The procedure is repeated for $N_S = 500$ simulations of the initial homogeneous region, estimating a mean value of $Z$ over $N$s for each ($\tau^d$, $k$) pair. For the region formed by 'homogenous' and 'discordant' sites, named as mixed region, $Z$ is expected to be larger for larger $k$ values. Indeed, a larger number of discordant sites in the region should increase the degree of regional heterogeneity. Also, $Z$ is expected to be larger as the difference between $\tau^R$ and $\tau^d$ gets larger, since the addition of sites with a 'larger discordance' should increase the degree of regional heterogeneity. On the other hand, for the sub-region formed by the sites belonging to the initial homogeneous region, $Z$ is expected to keep the same values regardless of the value of $k$, which in this case is the number of initial sites removed. The degree of regional heterogeneity should be relatively constant in this case, since all the sites belong to the same initial homogeneous region. Note that a mixed region can be seen as a sort of bimodal region used in other studies (e.g. Chebana and Ouarda, 2007).

## 3  Heterogeneity measures

The aim of this section is to present and develop heterogeneity measures based on different approaches to be assessed by the procedure proposed in Sect. 2. Heterogeneity measures are selected as a result of a general and comprehensive literature review in a number of fields including hydrology. We can distinguish three types of measures: (a) known in RHFA; (b) derived from recent approaches in RHFA; and (c) used in other fields and adapted here to the regional hydrological context. Therefore, a total of eight measures are considered.

### 3.1.  Measures known in RHFA

The first group consists of the well-known statistics $H$, $V$, $H_2$ and $V_2$ (Hosking, 2015), as well as the k-sample Anderson-Darling ($AD$) statistic (Scholz and Stephens, 1987; Scholz and Zhu, 2015).

Even though $H$ is not properly defined as a heterogeneity measure for ranking the degree of heterogeneity of several regions (see Sect. 1), it is considered in this study because it is commonly adopted in regional analysis. As the aim of this study is to provide a general heterogeneity measure, its associated distribution-free statistic $V$ is also considered. Specifically, $V$ is a statistic of the dispersion of the sample L-CV $t$ in a region, expressed as:

$$V = \sqrt{\frac{\sum_{i=1}^{N} n_i(t^i - t^R)^2}{\sum_{i=1}^{N} n_i}}, \tag{1}$$

with

$$t^R = \frac{\sum_{i=1}^{N} n_i t^i}{\sum_{i=1}^{N} n_i}, \tag{2}$$

where $t^i$ is the sample L-CV at site $i$ and $t^R$ is its associated regional average. $H$ is a measure of the variability of $t$ in the region compared with that expected for simulated homogeneous regions. It is built by normalising $V$ by its mean $\mu_V$ and standard deviation $\sigma_V$:

$$H = \frac{V - \mu_V}{\sigma_V},\tag{3}$$

where $\mu_V$ and $\sigma_V$ are obtained from $N_H = 500$ simulated homogeneous regions with the same $n$ and $N$ as the given region, generated from a kappa distribution fitted to the regional average L-moment ratios.

The extensions of $V$ and $H$ by considering not only $t$ but also the sample L-skewness coefficient $t_3$, traditionally known as $V_2$ and $H_2$, are also included in this study. Their inclusion is motivated by recent results regarding the usefulness of $H_2$ for testing homogeneity when considering different thresholds from those of $H$ (Wright et al., 2014):

$$V_2 = \frac{\sum_{i=1}^{N} n_i \sqrt{(t^i - t^R)^2 + \left(t_3^i - t_3^R\right)^2}}{\sum_{i=1}^{N} n_i},\tag{4}$$

$$H_2 = \frac{V_2 - \mu_{V_2}}{\sigma_{V_2}},\tag{5}$$

where $t_3^i$ is the sample L-skewness coefficient at site $i$ and $t_3^R$ is its associated regional average. $t_3^R$ is defined analogous to $t^R$ in Eq. (2). In order to avoid results conditioned on the given value of $t^R$ and $t_3^R$, $V$ and $V_2$ are standardised here by their regional values, defining $V'$ and $V_2'$ respectively as:

$$V' = \sqrt{\frac{\sum_{i=1}^{N} n_i \left(\frac{t^i - t^R}{t^R}\right)^2}{\sum_{i=1}^{N} n_i}},\tag{6}$$

$$V_2' = \frac{\sum_{i=1}^{N} n_i \sqrt{\left(\frac{t^i - t^R}{t^R}\right)^2 + \left(\frac{t_3^i - t_3^R}{t_3^R}\right)^2}}{\sum_{i=1}^{N} n_i},\tag{7}$$

The $AD$ statistic, which is a rank-based statistic based on comparing the at-site empirical distributions with the pooled empirical distribution of the data, is also included in this first group:

$$AD = \frac{1}{M} \sum_{i=1}^{N} \frac{1}{n_i} \sum_{j=1}^{M-1} \frac{\left(Mm_{ij} - jn_i\right)^2}{j(M - j)},\tag{8}$$

where $M = \sum_{i=1}^{N} n_i$ and $m_{ij}$ is the number of observations in the $i^{th}$ sample not greater than $y_j$, where $y_1 < \cdots < y_M$ is the pooled ordered sample of the data, which in the regional context entails considering the data of each site first divided by its corresponding mean and then ordered. The $AD$ statistic has already been considered in several studies. Viglione et al. (2007) assessed its behaviour as a homogeneity test statistic, recommending its use when $t_3^R > 0.23$. Wright et al. (2015) evaluated

its performance as a heterogeneity measure regarding its ability to be a surrogate of the quantile error, yet obtaining a weak performance partially attributed to a possible influence of the procedure used for estimating errors.

## 3.2. Measures derived from recent approaches in RHFA

The second group is represented by a measure derived from a relatively novel approach in which the confidence interval for the at-site L-CV $t^i$ (with $i$: 1,…$N$) is estimated and compared with $t^R$. The focus is to evaluate how often the latter is included in such confidence intervals in order to assess if differences between $t^i$ and $t^R$ can be attributed to sample variability or to regional heterogeneity.

Viglione (2010) proposed a procedure for obtaining the confidence interval for L-CV without considering a given parent distribution of the data, applying it to a didactic illustration for comparing several regional approaches. The procedure is summarised below: the variance of the sample L-CV $t$, var($t$), is estimated according to Elamir and Seheult (2004) which is implemented in Viglione (2014); simple empirical corrections are applied on $t$ and var($t$) based on the values of $t_3$ and $n$; and the confidence interval for $t$ is then obtained from a log-Student's distribution considering corrected values of $t$ and var($t$). For instance, for a 90% confidence interval, a region is considered as heterogeneous if $100 - (P_{05} + P_{95}) \ll 90\%$, where $P_{05}$ ($P_{95}$) is the percentage of sites for which $t^R$ is below (above) the confidence interval for $t^i$. The larger ($P_{05} + P_{95}$) is, the larger the regional heterogeneity will be. Das and Cunnane (2011) obtained such a confidence interval based on simulations from a GEV distribution, with the aim of evaluating if a usual method to select catchment descriptors for delineating regions in Ireland provided homogeneous regions. The number of sites for which $t^R$ is outside the $t^i$ confidence intervals is considered as a measure of heterogeneity, also expressed as a percentage of sites.

In the present study, the heterogeneity measure considered regarding this approach is named as $P_{\mathrm{CI}}$ and defined as the total percentage of sites in the region for which $t^R$ is outside the 90% confidence interval for $t^i$. As the parent distribution of the data is unknown in practice, such a confidence interval is estimated following the aforementioned distribution-free approach.

## 3.3. Measures used in other fields and adapted here to the regional hydrological context

The last group consists of the Gini index ($GI$) (Gini, 1912; Ceriani and Verme, 2012), which is a measure of inequality of incomes in a population commonly used in economics; and of a measure based on the entropy-based Kullback-Leibler ($KL$) divergence (Kullback and Leibler, 1951), which estimates the distance between two probability distributions and is used for different purposes in a number of fields including hydrology (e.g. Weijs et al., 2010).

The definition of the $GI$ is usually given according to the Lorenz curve (Gastwirth, 1972), but it can be expressed in other ways. Specifically, the sample $GI$:

$$GI = \frac{\sum_{i=1}^{n} \sum_{j=1}^{n} |x_i - x_j|}{2n^2\mu},$$ 

(9)

corrected for short sample sizes can be defined as (Glasser, 1962; Zeileis, 2014):

$$GI = \frac{\sum_{i=1}^{n}(2i - n - 1)x_{i:n}}{n(n-1)\mu}, \qquad (10)$$

where $x_{i:n}$ are the sample order statistics and $\mu$ is their mean. Theoretically, $GI$ ranges from zero to one. The former is obtained when all the $x_i$ values are equal, and the latter is given when all but one value equals zero (in an infinite population). Note that although $GI$ has not been directly applied to hydrology, it is connected with the well-known L-moments which do. Both are based on sample order statistics. Indeed, $GI = GMD/2\mu$ (for $\mu > 0$), where $GMD$ is the Gini's

mean difference statistic (Yitzhaki and Schechtman, 2012); and $GMD = 2l_2$, where $l_2$ is the second sample L-moment (Hosking and Wallis, 1997). Hence, $GI$ corrected for short samples corresponds to the sample L-CV $t$ (Hosking, 1990), which implies that if $GI$ is applied on the flood observations at site $i$, the result is $t^i$. Then, in order to adapt $GI$ to the regional hydrological context, in this study $GI$ is applied on $t^i$ over sites. This provides a value of the inequality or variability of the at-site L-CV $t^i$ in the region, and hence it can be seen as a measure of the heterogeneity of the region.

Therefore, the measure considered in this study is $GI(t^i, i = 1, ..., N)$:

$$GI = \frac{\sum_{i=1}^{N}(2i - N - 1)t_{i:n}}{N(N-1)\bar{t}}, \qquad (11)$$

where $t_{i:n}$ are the sample order statistics, $\bar{t}$ is their mean, and the number of sites $N$ corresponds to the data length of $t$. Note that $GI(t^i, i = 1, ..., N)$ is equivalent to $t(t^i, i = 1, ..., N)$. Also, note that this is somehow analogous to the approach based on moments used in early studies (e.g. Stedinger and Lu, 1995), where the coefficient of variation ($Cv = \sigma/\mu$) of the coefficient of variation of the data (i.e. $Cv(Cv^i, i = 1, ..., N)$) was used for building simulated regions; defining

homogeneous regions for $Cv(Cv^i) = 0$ and extremely heterogeneous regions for $Cv(Cv^i) \geq 0.4$.

The $KL$ divergence (so-called relative entropy) of the probability distribution $P$ with respect to $Q$ is defined as:

$$KL(P||Q) = \int p(x)\ln[p(x)/q(x)]\,dx \qquad (12)$$

where $p$ and $q$ are the density functions. The expression related to the discrete case is the following (e.g. Hausser and Strimmer, 2009)

$$KL(P||Q) = \sum_{m} P_m \ln\left(\frac{P_m}{Q_m}\right) \qquad (13)$$

for which nonparametric versions of the probabilities $P$ and $Q$ may be considered, such as a kernel density function, in order

to avoid subjectivity in selecting a given parametric probability distribution. $KL_{ij}$ can then be defined as the $KL$ divergence of the probability distribution at site $i$ with respect to the probability distribution at site $j$, where $KL_{ij} \neq KL_{ji}$. The dissimilarity matrix of the region is obtained by computing the $KL$ divergence between sites as:

$$D_{KL} = \begin{pmatrix} KL_{11} & \dots & KL_{1N} \\ \vdots & KL_{ij} & \vdots \\ KL_{N1} & \dots & KL_{NN} \end{pmatrix} \tag{14}$$

The degree of regional heterogeneity is then evaluated by $\|D_{KL}\|$, which in this study is considered as the absolute column sum normalized norm:

$$\|D_{KL}\| = \frac{max_j \sum_i |KL_{ij}|}{N} \tag{15}$$

## 4  Results

Simulation results obtained by the application of the proposed assessment procedure (Sect. 2) to the considered heterogeneity measures (Sect. 3) are presented in this section. Note that a summary of the results obtained from each step is presented in Table 1.

### 4.1.  Sensitivity analyses

Results of the effect of varying factors defining a region (Sect. 2.2) are presented through boxplots and mean values of the heterogeneity measure over $Ns = 500$ simulations of the corresponding region, in order to show complete information. Results for $\tau^R = 0.2$ refer to those related to the reference region. Figure 1 shows that all considered measures seem to be positively correlated with an increasing heterogeneity rate $\gamma$. This means that their behaviour is appropriate as they may indicate heterogeneity. This dependence is less pronounced for $H_2$ and $V_2'$, which are the measures that depend on both $t$ and $t_3$; and for $AD$ and $\|D_{KL}\|$, which are based on the whole information.

The effect of $N$ on the considered measures is shown for $\tau^R = 0.2$ when $\gamma = 0\%$ (i.e. 'perfect' homogeneous regions) and $\gamma = 50\%$ in Figs. 2 and 3, respectively. In both cases, it is found that $V'$, $V_2'$, $P_{CI}$ and $GI$ are not affected by $N$, although they show some departure from their constant $Z$ mean value and a larger variability (i.e. larger box) when $N \leq 10$. In this regard, Das and Cunnane (2012) also found an effect for $N < 10$ on quantile error measures (considering $n = 35$). In general this effect is less marked for $GI$ when $\gamma = 0\%$ (Fig. 2c,d) and for $V'$ and $V_2'$ when $\gamma = 50\%$ (Fig. 3a,b).

It is also found that results for $H$, and to a lower degree for $H_2$, change depending on the value of $\gamma$. These measures do not depend on $N$ for $\gamma = 0\%$ (Fig. 2a,b); yet they do for $\gamma = 50\%$ (Fig. 3a,b). This is likely due to the nature of $H$ and $H_2$ as homogeneity test statistics. Note that this undesirable effect increases as $\gamma$ increases (e.g. Fig. 4). $\|D_{KL}\|$ is affected by $N$ for both $\gamma = 0\%$ and $\gamma = 50\%$. The same holds for $AD$, for which such dependence is higher.

The influence of varying regional average L-moments is shown by comparing the $Z$ mean values for $\tau^R = 0.4$ with those previously obtained for $\tau^R = 0.2$. $Z$ mean values varying $\gamma$ are displayed in Fig. 1b,d. In this regard, $V_2'$ and $AD$ fail to compare regions with the same $\gamma$ but different $\tau^R$, as results for $\tau^R = 0.2$ and $\tau^R = 0.4$ are far from each other. Regarding $H$ and

$H_2$, this effect is worse for higher degrees of regional heterogeneity than for smaller ones; whereas $V'$, $GI$ and $\|D_{KL}\|$ show the opposite behaviour with an overall better performance of $V'$ and $GI$. $P_{CI}$ is better able to compare regions with either small or high $\gamma$. Results for $Z$ mean values varying $N$ are displayed for $\gamma = 0\%$ in Fig. 2b,d; and for $\gamma = 50\%$ in Fig. 3b,d. In both cases $V_2'$ and $AD$ fail to compare regions with the same $\gamma$ but different $N$. A suitable performance is found for $V'$, $P_{CI}$

and $GI$ for $\gamma = 50\%$; whereas a worse performance is found for $H$, $H_2$ and $\|D_{KL}\|$ (Fig. 3b,d). This performance of $H$, $H_2$ and $\|D_{KL}\|$ is also shown for $\gamma = 0\%$ (Fig. 2b,d), for which the remaining measures also present similar results. In this regard, it is important to remark that no 'perfect' homogeneous regions exist in reality (Stedinger and Lu, 1995). And that according to the practical threshold $H < 2$, commonly used for considering a region as homogeneous enough to perform a regional analysis, even regions with $\gamma = 50\%$ may be taken as homogenous in practice (see values of $H$ for $\gamma$ in Fig. 1a). Hence, for

the purpose of the assessment of the regional heterogeneity degree, the behaviour of the measures for $\gamma = 50\%$ is more relevant than for $\gamma = 0\%$.

Finally, the effect of varying the record length $n$ for $\gamma = 0\%$ and $\gamma = 50\%$ is shown in Fig. 5. Recall that it is expected that increasing $n$ affects $Z$, as more information of the at-site distributions is available in such a case. In this regard, it is found that the measures $H$, $H_2$, $AD$ and $P_{CI}$ are not (or slightly) affected by $n$ when $\gamma = 0\%$, but they highly increase their values as

$n$ increases when $\gamma = 50\%$. Whereas $V'$, $V_2'$, $GI$ and $\|D_{KL}\|$ are affected by $n$ when $\gamma = 0\%$; becoming less affected when $\gamma = 50\%$, by decreasing less their values as $n$ increases. As a result, $V'$ and $GI$ are the only measures that become relatively stable for a given data length. Such a data length is around $n = 30$, which is a value usually considered in practice (e.g. Hosking and Wallis, 1997, page 134; Chebana and Ouarda, 2009). It can be mentioned that for a very small data length ($n = 10$), the approximation used in $P_{CI}$ for estimating var($t$) was not reliable. Nevertheless, this issue is not relevant since such a data

length is too short to be considered in practice, and such values do not affect the overall interpretation of the results.

As a result of the aforementioned qualitative sensitivity analysis results (see Table 1 for a summary), $V'$, $P_{CI}$ and $GI$ are considered as potentially suitable heterogeneity measures. Thus, the following steps of the assessment procedure are only applied to these measures. Results of $H$ are also included for comparison purposes.

### 4.2. Success rate

The ability of the measures to identify the most heterogeneous region between two regions A and B is shown via the success rate $SR$ (Sect. 2.3). A summary of the results obtained for $\tau^A = \tau^B$ and $\tau^A \neq \tau^B$ (with $\gamma_A < \gamma_B$), when considering several values of $N$ and $\gamma$ for each region is displayed in Table 2 to facilitate their interpretation. Note that each combination $\tau^A$ vs. $\tau^B$ corresponds to a total of 48 cases obtained by varying $N$ and $\gamma$. Results for a small difference between $\tau^R$ values, characterised by $\tau^A = 0.2 \neq \tau^B = 0.3$ and vice versa; and for a large difference, characterised by $\tau^A = 0.1 \neq \tau^B = 0.4$ and

vice versa, are displayed as representative of the behaviour of the measures. Note that the summarised information reflects the main conclusions extracted from the partial results.

The *SR* average is shown as a notion of the overall behaviour of the measures. Recall that the larger *SR* is, the better *Z* will be. When $\tau^A = \tau^B$ the *SR* average of *H*, *V′* and *GI* are comparable, with *V′* and *GI* leading to the largest values; while $P_{CI}$ leads to the lowest ones. When $\tau^A < \tau^B$ the largest *SR* average is obtained for *V′* and is very closely followed by *GI*. Yet, in this case *H* presents a worse behaviour, which is similar to that of $P_{CI}$. When $\tau^A > \tau^B$ the situation changes, with *H* leading

to the largest values. Yet, the difference between the values obtained by *V′* (or *GI*) and *H* is less marked than when $\tau^A < \tau^B$. Note that the larger the difference between $\tau^A$ and $\tau^B$ is, the larger the difference between the *SR* average of *H* and *V′* (or *GI*) is; whereas the value of $P_{CI}$ remains almost constant. Therefore, although $P_{CI}$ does not obtain the greatest values in any case, it outperforms *H* or *GI* (and *V′*) when $\tau^A \ll \tau^B$ or when $\tau^A \gg \tau^B$, respectively, i.e. for high differences between $\tau^A$ and $\tau^B$. The best results for the total *SR* average are obtained by *GI*, followed by *V′*.

The *SR* minimum and *SR* maximum are displayed as a notion of the variability of the *SR* results (Table 2). Results related to the *SR* minimum are analogous to those obtained by the *SR* average; giving *H* an overall worse behaviour. This highlights the low ability of *H* to identify the most heterogeneous region in certain circumstances. Note that the overall behaviour of *H* regarding *SR* is partially due to existing trends regarding *N* and $\tau^R$. *H* obtains larger heterogeneity values as *N* increases and as $\tau^R$ decreases (as shown in Fig. 3b), entailing an 'unfounded' better behaviour when $\tau^A > \tau^B$ and $N_A < N_B$, and vice versa.

Also note that all measures have difficulties obtaining a large *SR* minimum when $\tau^A > \tau^B$. This includes *H* also, even though it obtained a good *SR* average in such a situation. This arises from the fact that, in such a case, the region with the lowest degree of heterogeneity (region A) is associated with a larger $\tau^R$ entailing a larger sample variability, and complicating its identification as the less heterogeneous region. *SR* maximum values show that even though the maximum difference between $\gamma_A$ and $\gamma_B$ considered in the analysis is 30%, all measures obtain (in certain circumstances) a *SR* equal or close to 100%. In

summary, *GI* obtains the best results for the *SR* analysis followed by *V′*.

### 4.3. Evolution of the variability of the *Z* average with respect to the degree of regional heterogeneity

The variability of the heterogeneity measures as a function of the degree of regional heterogeneity, represented by γ, is shown in Fig. 6. The boxplot of the 22 representative (mean over *Ns* = 500) values of *Z* obtained from cases in which a given region A and a given region B with the same γ but different characteristics are considered is shown for varying values of γ in

the *x*-axis (see Sect. 2.4). As expected from the results of Fig. 1, heterogeneity measures in Fig. 6 increase with γ, showing a monotonic positive dependence. Regarding their variability along such a monotonic relation, *H* presents a different behaviour from the rest of the measures. It shows a strong increasing variability as γ increases. Then, in this case, *H* overlaps its interquantile ranges from γ = 70% to 100%. This behaviour may imply an unappropriated ranking of the regions with these high values of the heterogeneity rate γ. Indeed, overlapped values cannot be considered significantly different, whereas

they correspond to two different γ values. Such behaviour is not seen for the other considered measures. In this regard, an overall favorable larger distance between interquantile ranges is found for *V′*, followed by $P_{CI}$ and then *GI*. However, the four considered measures present an overlapping for γ = 0% and 10%. This may imply an unappropriated ranking of the

regions related to these very small values of $\gamma$, yet those regions are less common in practice. In summary, $V'$ obtains the best performance for the variability evolution analysis. It presents a small variability for a given $\gamma$ value; and it almost presents no overlapping between interquantile ranges for varying $\gamma$.

### 4.4. Effect of discordant sites

The effect of discordant sites (Sect. 2.5) is shown in Fig. 7. The mean values of the heterogeneity measures over $Ns = 500$ are obtained when replacing $k$ sites (with $k = 1,\dots, 10$) in an initially homogeneous region (with $N = 20$) by $k$ discordant sites belonging to a given parent distribution defined by $\tau^d$. Note that unlike Fig. 1, where the heterogeneity value of two kinds of regions with the same degree of heterogeneity but different regional L-CV may be compared; in Fig. 7 progressive changes in the heterogeneity of a single homogeneous region are assessed. For the mixed region formed by sites from both $\tau^R$ and $\tau^d$,

the overall results confirm that the considered measures involve larger values of $Z$ for larger $k$ values, as a result of replacing a larger number of discordant sites in the region; and larger values of $Z$ as the difference between $\tau^R$ and $\tau^d$ increases, as a result of replacing sites with a larger discordance (Fig. 7a).

However, when $\tau^d > \tau^R$ (Fig. 7b) the measures face some difficulties in ranking the degree of heterogeneity for high values of $k$. This is due to the larger sample variability entailed by the discordant sites in such a case, which makes the whole mixed

region seem less heterogeneous. Note that this is also the reason of the lack of asymmetry of the results regarding the vertical line at the midpoint of the $x$-axis (i.e. $\tau^d = 0.25 = \tau^R$). Nevertheless, not all measures are equally affected by this issue. $GI$ obtains the best results, as for instance it is able to differentiate the degree of heterogeneity for $k \leq 8$ when $\tau^d = 0.35$ and 0.4. It is followed by $P_{\mathrm{CI}}$, which behaves properly for $k \leq 8$ when $\tau^d = 0.35$ and for $k \leq 7$ when $\tau^d = 0.4$; and by $V'$, which obtains adequate results for $k \leq 7$ when $\tau^d = 0.35$ and $k \leq 6$ when $\tau^d = 0.4$. The worst results are obtained by $H$, which only behaves

properly for $k \leq 6$ when $\tau^d = 0.35$ and $k \leq 4$ when $\tau^d = 0.4$. Results for the sub-region formed by the remaining $(N - k)$ sites of the initial homogeneous region (Fig. 7a) support the results in Fig. 2, as $H$ and $GI$ are practically not affected by the number of sites of the homogeneous region, while $V'$ and $P_{\mathrm{CI}}$ present a slight decrease in their heterogeneity values as the number of sites $(N - k)$ decreases. In summary, $GI$ presents the best results for the analysis of discordant sites.

### 5 Discussion

Overall, $GI$ can be considered as the best heterogeneity measure among all the evaluated measures, closely followed by $V'$ (see a summary in Table 1). However, as expected, none of the measures are perfect, due to their inability to perfectly fulfill all the desirable properties in practice. $GI$ presents the advantage of being computed as a measure of the standardised mean distance between pairs of $t^i$ values. Hence, it does not depend on any assumptions concerning parameters or parent distributions. $V'$ is similar but it specifically depends on the estimate of the regional average $t^R$, as it compares it to each $t^i$

value. Thus, due to the similar but slightly better results obtained by *GI* and its widely accepted use in other fields, the use of *GI* would be preferable in practice.

*H* is by nature the statistic of a homogeneity test. Hence, it is defined to identify whether a given region can be considered as homogeneous or not, not to compare the heterogeneity degree of several regions. Note that this is also valid for other test statistics (e.g. *AD*). As a consequence of the intrinsic disadvantages of *H* (see Sect. 1) and the obtained results, the use of *H* as a heterogeneity measure for ranking regions is not recommended. The unsatisfactory results obtained for $V_2'$ and $H_2$ could be related to the way in which $t$ and $t_3$ are combined (see Sect. 3), which may not be appropriate for assessing the degree of regional heterogeneity. The unsuitable results associated with $\|D_{KL}\|$ could be related to considering the whole information of the data, which may mask the effect of factors favouring heterogeneity. It should be noted that other norms aside from the one in Eq. (15) were considered, but they did not lead to better performances. Further research should focus on the development of a better adaptation of the entropy-based measures to estimate the degree of regional heterogeneity.

The $P_{CI}$ measure is obtained without assuming a given parent distribution of the data; although it considers a log-Student distribution for estimating the L-CV confidence interval. Also, even though it depends partially on the selected confidence level, mean $P_{CI}$ values over $Ns = 500$ for different confidence levels (90% and 95%) were found to be highly correlated (not shown). This fact removes subjectivity from the use of $P_{CI}$ as a heterogeneity measure, as for such a purpose only the ranking of values is needed. It is also important to highlight the stable performance of $P_{CI}$ regardless of the value of $\tau^R$. This makes $P_{CI}$ outperform *GI* and $V'$ for identifying the most heterogeneous region when such a region has a much lower $\tau^R$ than others to be compared with (see Table 2). As a consequence, $P_{CI}$ and *GI* could be used together in practice as two different and complementary criteria. This is common in other applications; for instance several criteria are commonly applied when ranking candidate distributions (e.g. the Akaike information criterion and the Bayesian information criterion). It is important to mention that the use of $P_{CI}$ as a homogeneity test in practice may lead to the false rejection of homogeneous regions. Indeed, even when a region is 'perfectly' homogeneous ($\gamma = 0\%$) the mean value of $P_{CI}$ may indicate slight heterogeneity (e.g. it is slightly larger than 10% in Fig. 1).

As indicated in Sect. 1, the heterogeneity measures selected in this study may be used for the assessment of the degree of heterogeneity of regions obtained through the use of different delineation methods. When a region is divided into several sub-regions by a given delineation method, the *GI* (or $P_{CI}$) value can be evaluated at each sub-region. Then, the average value can be used to compare several delineation methods applied on the given region. The best delineation method will be the one with the lowest *GI* (or $P_{CI}$) value for the region of study (see Sect. 6 for an illustrative application). It is important to note that a heterogeneity measure should not be used as a decision variable for the delineation of regions, as it would imply using redundant information at different steps of the regional analysis. The heterogeneity measure can also be used for evaluating the heterogeneity of a given region when particular sites are removed, with the aim of helping in the identification of homogeneous regions. For instance, if a region is found as heterogeneous by using a given test and by entailing a number of discordant sites, the heterogeneity measure can help in the identification of the 'most homogeneous region' as a result of removing different combinations of sites. However, it is important to highlight that physical reasoning has to be provided for

removing a given (discordant) site. Thus the heterogeneity measure serves only as a facilitator for the identification of the site(s) to be further analysed (e.g. Viglione, 2010; Ilorme and Griffis, 2013).

## 6 Illustrative application

An illustrative application on observed data is presented for didactical purposes. The considered case study consists of 44
sites from the hydrometric station network of the southern part of the province of Quebec, Canada (for more description of the data and the region see Chokmani and Ouarda, 2004). The flow data are managed by the Ministry of the Environment of Quebec Services. Descriptors and at-site spring flood quantiles are available for the considered sites (Kouider et al., 2002). A summary of the statistics associated with spring maximum peak flow data, relevant descriptors for flood frequency analysis (e.g. Shu and Ouarda, 2007) and at-site spring flood quantiles is shown in Table 3. Note that due to the data used in this
application are observed instead of simulated, the real degree of heterogeneity of the regions, as well as the real parent distribution of the data are unknown. Thus, it is not possible to truly compare the performance of the different heterogeneity measures. In this regard, it is important to remark that the purpose of this illustrative application is then to show that commonly used criteria for identifying the best method for delineating regions may be subjective, as well as to guide practitioners in the use of heterogeneity measures.
The heterogeneity of the whole study region is evaluated by using a homogeneity test (Hosking and Wallis, 1997), resulting in a heterogeneous region ($H > 2$, see Table 4). Hence, the region is then divided into sub-regions by using cluster analysis (e.g. Burn, 1989) with the Ward's method, as it is one of the most applied in hydrology (e.g. Hosking and Wallis, 1997; Mishra et al., 2008). Because of the illustrative character of this application, three simple clustering settings are considered as the different delineation methods. Clustering A consists in applying cluster analysis based on catchment area, annual
mean total precipitation and annual mean degree-days below 0°C (see Table 3). Clustering B applies it only based on catchment mean slope; and Clustering C applies it based on catchment slope and fraction of the catchment controlled by lakes (see Table 3).

The results obtained by applying the best heterogeneity measure found in the present study, the $GI$, are shown in Table 4. For comparison purposes, the results obtained by applying commonly used criteria for identifying the best delineation method are also shown. They are $H$, and the quantile error calculated as the relative root mean square error (RRMSE) (see Sect. 1).
No more results are shown for space reasons and simplicity. Remind that the lower the heterogeneity measure or RRMSE value is, the better the delineation method will be. Note that in this case study, identifying the best delineation method implies identifying the best clustering setting. Two distributions commonly used in regional flood frequency analysis are considered when applying the index-flood method (Dalrymple, 1960) for estimating the quantiles to be evaluated through the
RRMSE. These distributions are the GEV and the generalised logistic (GLO) distribution; and the quantiles to be evaluated are the 10- and 100-year return period ($T$) spring flood quantile. RRMSE results related to a given distribution are only

shown in Table 4 if the regional distribution is accepted by a goodness-of-fit measure (Hosking, 2015). For comparison purposes, RRMSE results are obtained even if the given region is not "homogeneous" according to the homogeneity test.

According to the results in Table 4, $H$ average identifies Clustering B as the best delineation method. Nevertheless, due to $H$ is based on simulations, the $H$ value for the sub-regions slightly change if the procedure is repeated. In this particular case study, this implies that in some cases $H$ average in Clustering B becomes larger than $H$ average in Clustering A, and then Clustering A may be selected as the best one. Moreover, although not happening in this case study, it may occur that $H$ has negative values which may also complicate the evaluation of its average.

RRMSE average for $T = 100$ identifies Clustering A as the best delineation method. However, RRMSE average for $T = 10$ identifies Clustering C as the best one. Hence, a different decision is taken depending on the quantile considered for the assessment. Besides, it is also relevant to indicate that the selection of the best delineation method based on RRMSE may also depend on the regional distribution used. For instance, different distributions could be accepted for a given sub-region, resulting in different RRMSE values which could affect the final decision. In this regard, it is important to remark that when observed data are used, it is not possible to know neither the real regional parent distribution of the data, nor the real parent distribution to be used in obtaining the at-site quantiles used for evaluating RRMSE.

In the present application, the $GI$ identifies Clustering A as the best delineation method. The $GI$ seems to be a more objective criterion for identifying the heterogeneity of a region than criteria commonly used in practice. Besides, its use as heterogeneity measure is supported by the four-step simulation-based assessment procedure performed in the present paper. It is worth mentioning that Clustering A could be ideally assumed to be the best setting for forming sub-regions, as it is based on relevant descriptors for flood frequency analysis. However, this would just be an assumption that cannot be verified due to the use of observed data.

## 7    Conclusions

Delineation of homogeneous regions is required for the application of regional frequency analysis methods such as the index flood procedure. The availability of an estimate of the degree of heterogeneity of these delineated regions is necessary in order to compare the performances of different delineation methods or to evaluate the impact of including particular sites. Due to the unavailability of a well-justified and generally recognised measure for performing such comparisons, a number of studies have relied on measures that are not well-defined or approaches that involve additional steps during the delineation stage of regional frequency analysis.

In the present paper, a simulation-based general framework is presented for assessing the performance of potential heterogeneity measures in the field of regional hydrological frequency analysis (RHFA), according to a number of desirable properties. The proposed four-step assessment procedure consists of: sensitivity analysis by varying the factors of a region; evaluation of the success rate for identification of the most heterogeneous region; estimation of the evolution of the variability for the heterogeneity measure average with respect to the degree of regional heterogeneity; and study of the effect

of discordant sites. The procedure is applied on a set of measures including commonly used ones, measures that are derived from recent approaches, and measures that are adapted from other fields to the regional hydrological context. The assumption-free Gini Index ($GI$) frequently considered in economics and applied here on the L-variation coefficient (L-CV) of the regional sites obtained the best results. A lower performance was obtained for the measure of the percentage of sites ($P_{CI}$) for which the regional L-CV is outside the confidence interval for the at-site L-CV. However, this measure was considered relevant because of its stable behaviour regardless of the regional value of L-CV. The application of both measures may be recommended in practice. The use of different criteria to determine the degree of regional heterogeneity can help adequately identify the sites to be further analysed for obtaining homogeneous regions. Further research efforts are necessary to develop robust and general heterogeneity measures in the field of RHFA. In this study, an illustrative application is also included for didactical purposes. The subjectivity related to commonly used criteria in assessing the performance of different delineation methods is underlined through it. In this regard, further research may also focus on the application of heterogeneity measures to a variety of case studies in order to analyse practical aspects.

## Acknowledgements

The financial support provided by the Natural Sciences and Engineering Research Council of Canada (NSERC) and the Merit scholarship program for foreign students – Postdoctoral research fellowship of the *Ministère de l'Éducation et de l'Enseignement Supérieur du Québec* managed by the *Fonds de recherche du Québec – Nature et technologies* is gratefully acknowledged. The authors are also grateful to the editor S. Archfield, and to the reviewers W. Farmer, C. Rojas-Serna and K. Sawicz as well as two other anonymous reviewers whose comments helped improve the quality of this manuscript.

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

**Table 1. Summary of the results of the studied measures for the four-step assessment procedure. The behaviour of a given measure for each sensitivity analysis in step (i) is graded as: good (G), acceptable (A), bad (B) or unacceptable (U). Measures entailing an 'unacceptable (U)' behaviour are not assessed by the rest of steps; yet a complete assessment of $H$ is performed for comparison purposes. For steps (ii), (iii) and (iv) considered measures are ranked from the best results (1$^{st}$) to the worst results (4$^{th}$).**

| Measures | (i) Sensitivity analyses | | | | | (ii) Success rate (SR) | (iii) Variability evolution | (iv) Effect of discordant sites |
|---|---|---|---|---|---|---|---|---|
| | $\gamma$ | $N$ | | $\tau^R$ | $n$ | | | |
| | | $\gamma = 0\%$ | $\gamma = 50\%$ | | | | | |
| $H$ | G | G | U | B | B | 3$^{rd}$ * | 4$^{th}$ *** | 4$^{th}$ |
| $H_2$ | A | G | U | B | B | - | - | - |
| $V'$ | G | A | G | A | A | 2$^{nd}$ | 1$^{st}$ | 3$^{rd}$ |
| $V_2'$ | A | A | G | U | B | - | - | - |
| $AD$ | A | U | U | U | B | - | - | - |
| $P_{CI}$ | G | A | A | A | B | 4$^{th}$ ** | 2$^{nd}$ | 2$^{nd}$ |
| $GI$ | G | G | A | A | A | 1$^{st}$ | 3$^{rd}$ | 1$^{st}$ |
| $\|D_{KL}\|$ | A | U | U | B | B | - | - | - |

(*) High limitations for given circumstances; (**) Favorable stable values regardless of $\tau^R$ ; (***) Unacceptable results.

**Table 2. Summary of the success rate (*SR*) minimum, average and maximum of the considered measures (*H*, *V′*, $P_{CI}$ and *GI*), expressed in percentage, when comparing the heterogeneity of two regions A and B. For a given $\tau^A$ and $\tau^B$, such values are computed as the minimum, average and maximum of *SR* over 48 cases, respectively. For each case, *SR* is obtained as the mean over $N_s = 500$ simulations of two regions with $n = 30$ and given $N_A, N_B, \gamma_A$ and $\gamma_B$. Values in bold indicate the measure obtaining the largest *SR* minimum, *SR* average and *SR* maximum, respectively.**

| $\tau^A$ vs. $\tau^B$ | $\tau^A$ | $\tau^B$ | Minimum | | | | Average | | | | Maximum | | | |
|---|---|---|---|---|---|---|---|---|---|---|---|---|---|---|
| | | | *H* | *V′* | $P_{CI}$ | *GI* | *H* | *V′* | $P_{CI}$ | *GI* | *H* | *V′* | $P_{CI}$ | *GI* |
| $\tau^A = \tau^B$ | 0.2 | 0.2 | 33 | 47 | 40 | **50** | 74.5 | **77.9** | 67.3 | 77.7 | 99 | 99 | 91 | **100** |
| | 0.3 | 0.3 | 36 | 46 | 34 | **51** | 72.2 | 74.4 | 65.1 | **75.2** | **98** | 94 | 87 | **98** |
| $\tau^A < \tau^B$ | 0.1 | 0.4 | 7 | **69** | 36 | 57 | 58.8 | **86.4** | 61.4 | 85.7 | 87 | **98** | 83 | **98** |
| | 0.2 | 0.3 | 24 | 59 | 40 | **62** | 68.1 | **81.0** | 64.1 | 80.8 | 96 | 97 | 88 | **98** |
| $\tau^A > \tau^B$ | 0.3 | 0.2 | **47** | 34 | 33 | 34 | **77.3** | 70.4 | 67.6 | 71.8 | **100** | 96 | 92 | 97 |
| | 0.4 | 0.1 | **33** | 14 | 26 | 15 | **80.5** | 61.0 | 69.1 | 63.3 | **100** | 94 | 95 | 99 |
| Total average: | | | 30 | **45** | 35 | **45** | 71.9 | 75.2 | 65.8 | **75.7** | 97 | 96 | 89 | **98** |

**Table 3. Summary of the statistics of descriptors, spring maximum peak flow series, and available at-site quantiles for the 44 sites considered in the illustrative application.**

| Variables | | Unit | Min | Mean | Max | Std. |
|---|---|---|---|---|---|---|
| Descriptors | Catchment area | $km^2$ | 208 | 1062 | 5820 | 1075 |
| | Catchment mean slope | % | 0.99 | 2.67 | 6.81 | 1.29 |
| | Fraction of the catchment controlled by lakes | % | 0.1 | 1.63 | 5 | 1.38 |
| | Annual mean total precipitation | mm | 932 | 1057 | 1195 | 62 |
| | Annual men degree-days below 0°C | degree-day | 8589 | 11769 | 14158 | 1432 |
| Spring maximum peak flow series | Data length | years | 15 | 36 | 80 | 16.7 |
| | At-site mean | $m^3s^{-1}$ | 46.7 | 235.1 | 1137.4 | 209.7 |
| | At-site L-CV ($t$) | | 0.145 | 0.199 | 0.319 | 0.036 |
| | At-site L-skewness ($t3$) | | -0.032 | 0.139 | 0.404 | 0.098 |
| At-site quantiles | 10- year spring flood quantile | $m^3s^{-1}$ | 70.8 | 342.49 | 1616.08 | 298.93 |
| | 100-year spring flood quantile | $m^3s^{-1}$ | 107.8 | 469.11 | 2006.38 | 375.52 |

**Table 4. Results of the illustrative application: heterogeneity measures *H* and *GI*, and RRMSE. RRMSE values are associated with the GLO regional distribution; RRMSE values within parenthesis are associated with the GEV regional distribution. Bold values indicate the best result for each criterion.**

| Clustering | Sub-region | Nbr sites | *H* Value | *H* Average | *GI* Value | *GI* Average | RRMSE (%) $T=100$ Value | RRMSE (%) $T=100$ Average | RRMSE (%) $T=10$ Value | RRMSE (%) $T=10$ Average |
|---|---|---|---|---|---|---|---|---|---|---|
| Whole region | | 44 | 2.21 | 2.21 | 0.092 | 0.092 | 17.81 | 17.81 | 5.81 | 5.81 |
| A | A1 | 31 | 2.06 | 1.23 | 0.101 | **0.087** | 18.03 | **14.94** | 6.26 | 5.64 |
| A | A2 | 13 | 0.4 | | 0.074 | | (11.85) | | (5.01) | |
| B | B1 | 25 | 0.49 | **1.21** | 0.077 | 0.094 | 16.34 | 18.26 | 5.22 | 5.93 |
| B | B2 | 19 | 1.93 | | 0.111 | | 20.18 | | 6.63 | |
| C | C1 | 26 | 2.05 | 1.39 | 0.100 | 0.091 | 18.46 | 16.92 | 6.22 | **5.48** |
| C | C2 | 18 | 0.73 | | 0.082 | | (15.38) | | (4.74) | |

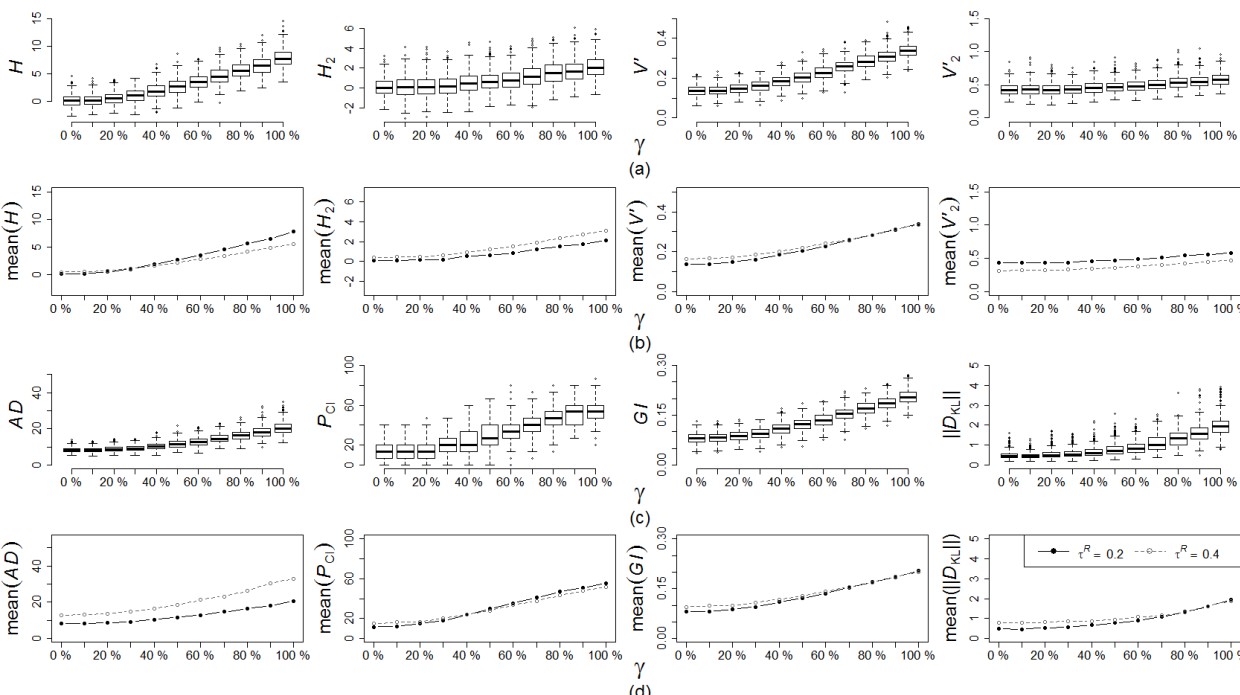

**Fig. 1. Sensitivity analysis: (a) (c) boxplots of the heterogeneity measures for $Ns = 500$ simulations of the reference region ($N = 15$, $n = 30$ and $\tau^R = 0.2$) varying the heterogeneity rate $\gamma$; and (b) (d) comparison of the corresponding mean with the one obtained by considering $\tau^R = 0.4$.**

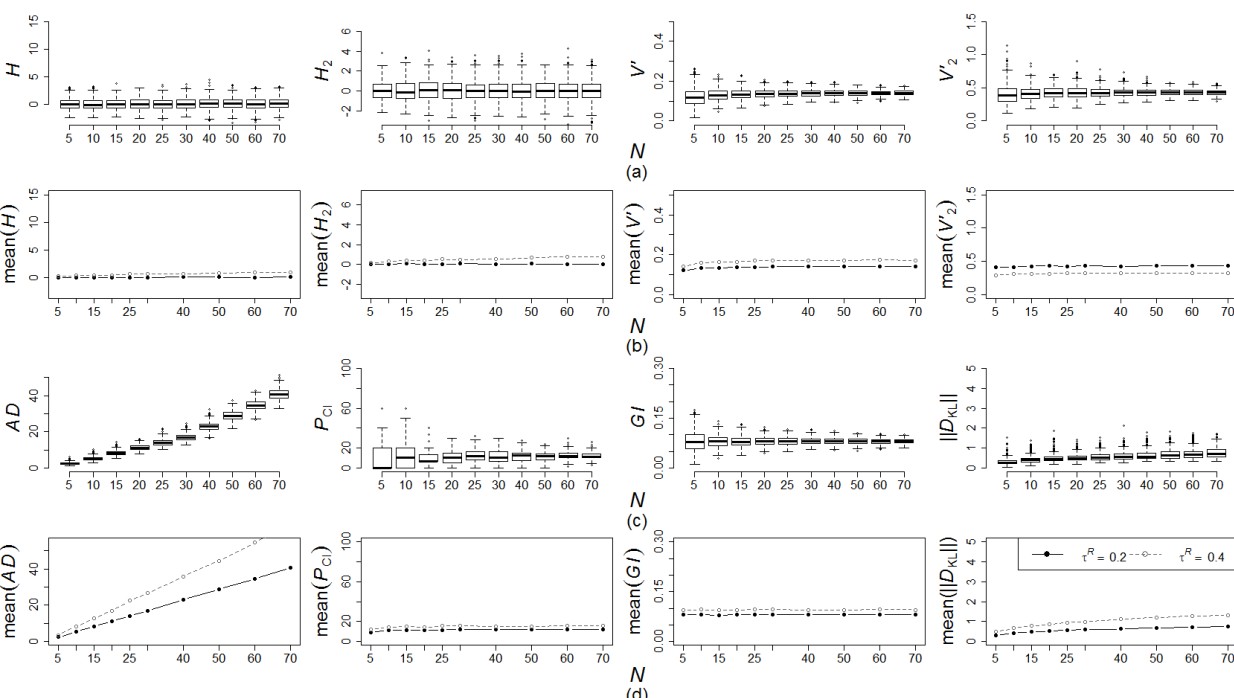

**Fig. 2. Sensitivity analysis: (a) (c) boxplots of the heterogeneity measures for $N$s = 500 simulations of the reference region ($n = 30$ and $\tau^R = 0.2$), with a heterogeneity rate $\gamma = 0\%$, varying the number of sites $N$; and (b) (d) comparison of the corresponding mean with the one obtained by considering $\tau^R = 0.4$.**

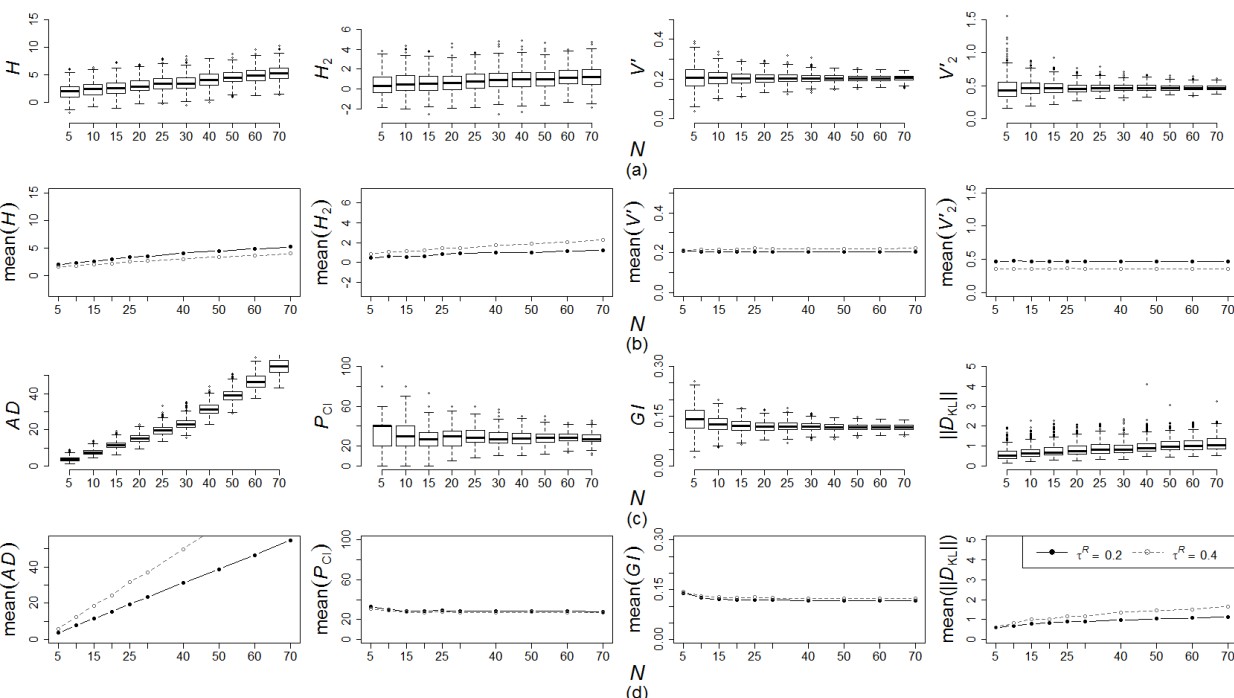

**Fig. 3.** Sensitivity analysis: (a) (c) boxplots of the heterogeneity measures for $Ns = 500$ simulations of the reference region ($n = 30$ and $\tau^R = 0.2$), with a heterogeneity rate $\gamma = 50\%$, varying the number of sites $N$; and (b) (d) comparison of the corresponding mean with the one obtained by considering $\tau^R = 0.4$.

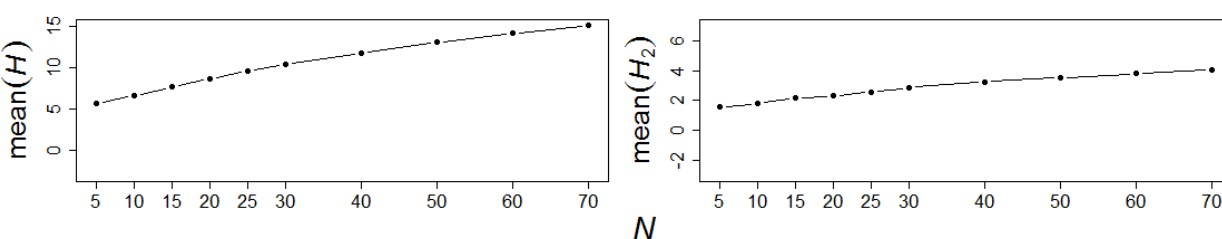

**Fig. 4.** Sensitivity analysis: mean of $H$ and $H_2$ over $Ns = 500$ simulations of the reference region ($n = 30$ and $\tau^R = 0.2$) for a heterogeneity rate $\gamma = 100\%$, varying the number of sites $N$.

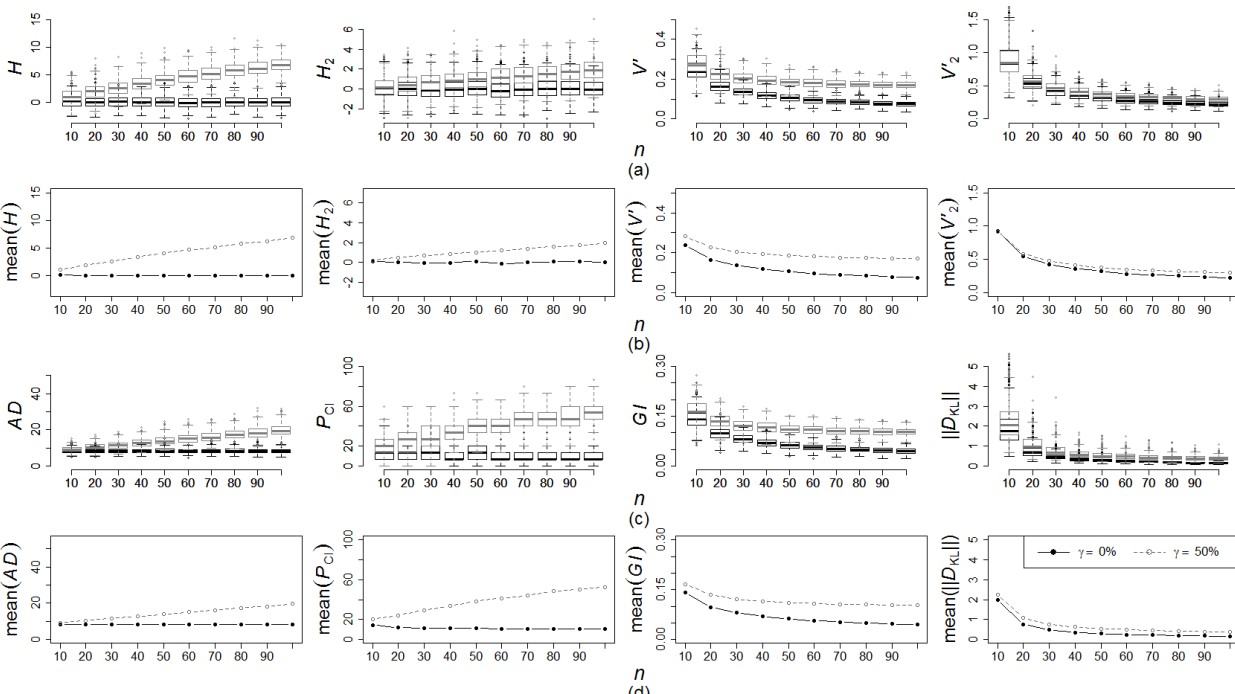

**Fig. 5. Sensitivity analysis: (a) (c) boxplots of the heterogeneity measures for $Ns = 500$ simulations of the reference region ($N = 15$ and $\tau^R = 0.2$), for a heterogeneity rate $\gamma = 0\%$ and $\gamma = 50\%$, varying the data length $n$; and (b) (d) comparison of the corresponding mean for $\gamma = 0\%$ and $\gamma = 50\%$.**

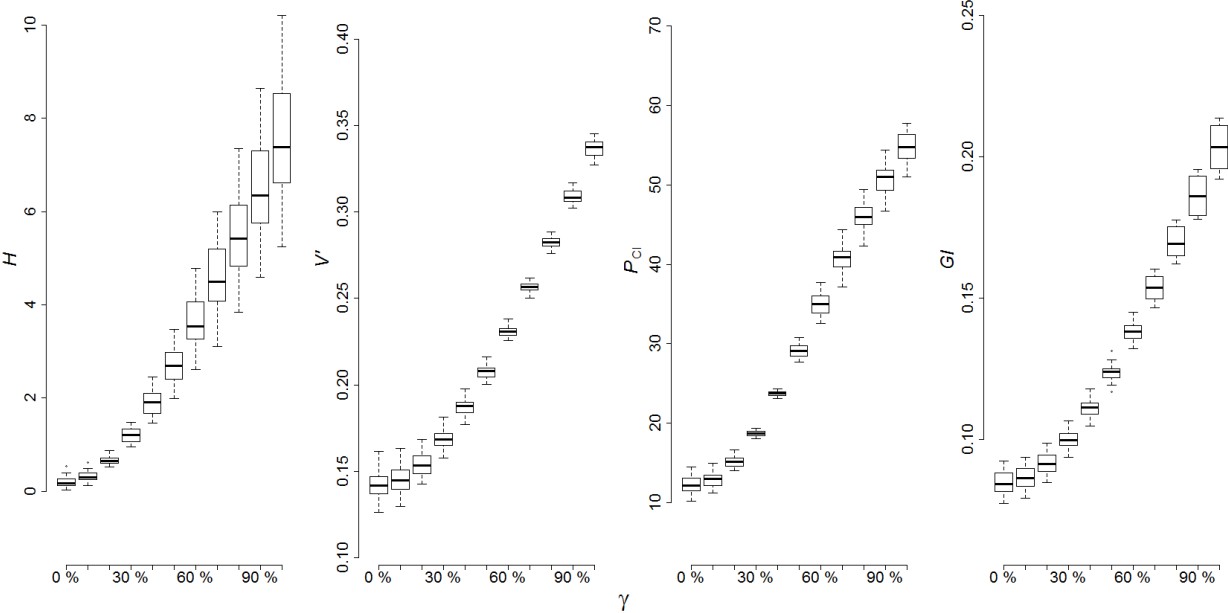

**Fig. 6. Boxplots of representative values of the heterogeneity measure average obtained for 22 cases, varying the heterogeneity rate γ in the *x*-axis. For each case, such a representative value is obtained as the average between a given region A and a given region B over $N$s = 500 simulations of the given regions, entailing the same γ (i.e. $\gamma_A = \gamma_B$) but different characteristics (i.e. $N_A \neq N_B$ or $N_A = N_B$ with $\tau^A \neq \tau^B$).**

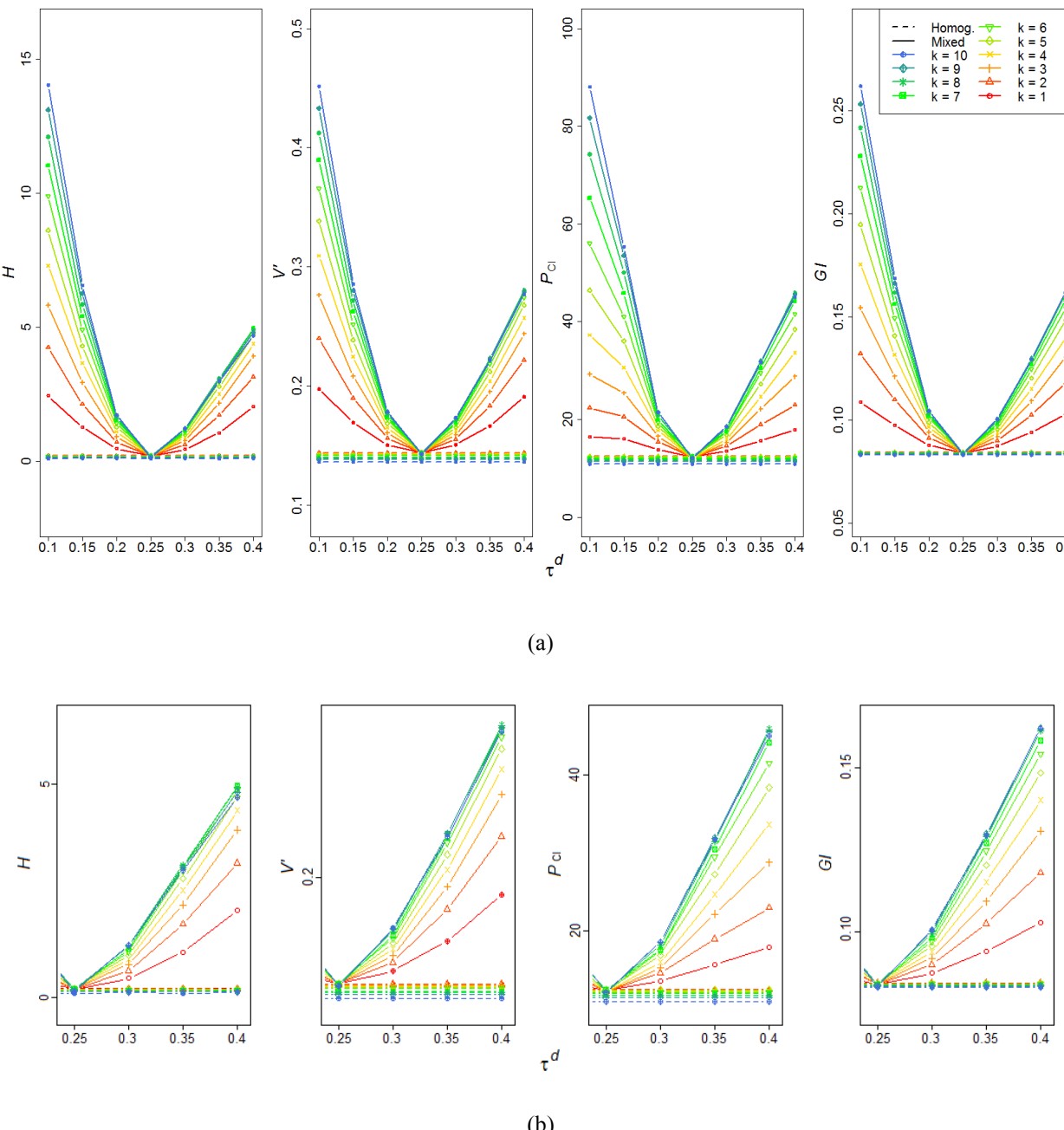

(a)

(b)

5 **Fig. 7. Mean values of the heterogeneity measures over $Ns$ = 500 simulations of a given homogeneous region with $N$ = 20 sites, $n$ = 30 and $\tau^R$ = 0.25, for which $k$ sites are replaced by $k$ discordant sites generated by a GEV with L-Cv $\tau^d$, varying $\tau^d$ in the $x$-axis: (a) full plot; and (b) zoom to the right part of the $x$-axis.**