# Peer review of "Heterogeneity measures in hydrological frequency analysis: review and new developments"

_Hydrology and Earth System Sciences, 2016_

## Referee Comment (RC1) · C. Rojas-Serna (Referee) · 7 Jun 2016

It's a good idea to determine the homogeneous zones from the identification of the heterogeneity between both, nevertheless, the amount of data used in this work is poor. This has consequences in the discussion of results because is not clear which is the hydrological variable that determines the heterogeneity. My comment is focused in the data used in this work: There is not a section where data used to apply the methodology are presented This limits the understanding of the heterogeneity in hydrology. It is unclear whether this is a watershed or region where the variable n is gauging stations. Or maybe n may be the number of meteorological stations. In that case the heterogeneity would be defined by climatic characteristics. I suggest include a section where the study area and the hydrological variable that is considered to define heterogeneity will be presented. So also describe the importance of data in the discussion of the

results.

---

## Referee Comment (RC2) · Anonymous Referee #2 · 13 Jul 2016

**GENERAL COMMENTS**

I really enjoyed reading this paper. I appreciate the efforts taken by the authors in contributing to the field of Regional Frequency analysis (RFA) by proposing a new index to assess heterogeneity of a region. There is clarity in describing the assumptions/drawbacks associated with the existing heterogeneity measures. Also, it is nicely stated why there is a need for new heterogeneity measures in RFA. The criteria that are defined to compare various heterogeneity measures with a new measure look adequate.

**SPECIFIC COMMENTS**

1. In the introduction section, literature review regarding regional hydrological frequency analysis is not complete.

[Figure]

2. Page 3, Line 32, "…focused on the delineation step". What do you mean by "..focussed on delineation step"? How Gini Index (proposed in this paper) account for the delineation step?

3. Page 4, line 28-29; what is the reason behind considering a linear relationship for L-CV and why gamma/2 was used?

4. Page 10, line 25-26; Gini index considered in this study is a function L-CV . Can GI be expressed in terms of L-skewness coefficient on the similar line as it was expressed in terms of L-CV? If yes, then will there be any change in the overall results from this study? I am asking this question because L-skewness would be more uncertain for observed data (due to involvement of higher (third) moment) which can influence the heterogeneity measure H2 or V2 (e.g variation in (V2) H2 heterogeneity measure would be more as compared to (V1) H1 heterogeneity measure; even visible in Figures 1, 2 and 3). As the indices are compared based on hypothetical regions (using Monte Carlo simulation), the effect on L-skewness coefficient may not be noticeable. But in practice, there can be uncertainty associated with the estimation of L-skewness coefficient.

5. What is the range GI index?

6. Page 16, line 3-10 describes use of GI in assessing the delineation methods. The methodology discussed is not clear. Different delineation methods have different criteria to select the optimal delineation/regionalization solution (e.g. AIC or BIC, Davies-Bouldin index in the case of hard clustering algorithms, Xie-Beni in the case of fuzzy clustering algorithm). Are you trying to say that we can rank the delineation methods based on GI index value to arrive at best delineation method for regionalization which then can be used to perform regional frequency analysis? If yes, then I think this kind of approach is even possible with any other Index (e.g., H1, H2). Authors have mentioned on Page 3, line 32 that the proposed index (e.g., GI) can alleviate the drawback associated with delineation method as compared to the conventional heterogeneity indices (H1, H2 etc).

7. In Figure 7, impact of heterogeneity due to addition of discordant sites to homogeneous regions is presented. A similar observation can be concluded from Figures 1 where the indices are assessed with increase in the heterogeneity percent (gamma value). In the case of Figure 1, mean of conventional heterogeneity measures (H1, H2, AD) tend to diverge with increase in the degree of heterogeneity while in Figure 7, Heterogeneity measures tend to converge (coming close) with increase in the number of discordant sites (i.e. increase in the degree of heterogeneity). Kindly clarify this point. However, in the case of GI, in both figures1 and 7, convergence is visible. Why is this happening?

8. The results obtained from the Monte Carlo simulation study are encouraging. Only thing missing from the paper is to perform analysis using observed data which would help to strengthen the conclusions drawn from the study. Analysis based on observed data would provide more clarity on the aforementioned concerns especially on Specific comment 4.

TECHNICAL CORRECTIONS

1. First line of Abstract reads "Regional frequency analysis is needed to estimate hydrological quantiles at ungauged sites or to improve estimates at sites with short record lengths, by transferring information from gauged sites." I think the regional transfer of information is possible with hydrologically similar sites only. Hence, "...hydrologically similar gauged sites" instead of "...gauged sites" looks more appropriate.

---

## Referee Comment (RC3) · Anonymous Referee #3 · 16 Jul 2016

The manuscript titled 'Heterogeneity measures in hydrological studies: review and new developments' presents a summary of the state current state of Regional Hydrologic Frequency Analysis (RHFA). Generally, I found the manuscript to be a very interesting and information dense product that I enjoyed reading. However, I think that there are missing components that limited my understanding of the implications of this study. This manuscript has a lot of ground to cover to get to it's results, and I encourage the authors to include key information and reorganize some of the sections as per my general comments below.

I was not able to find the data source in which the study was applied. It seems as though the data might be synthetic and generated as a hypothetical, but that is not clear. Please include at lease a small section specifically about how the data was used (if measured from real data) or synthesized (if it was generated by the authors). Please

[Figure]

include this data, or summary of data, either in the manuscript itself or as supplemental material.

The Gini Index is a very popular index to determine economic equality as the authors mention, but there should be additional descriptions about why the Gini Index was applied in the way that it was. Many of the other methods have been used in the past and are presented as benchmarks. However, the Gini Index is fairly new in hydrologic studies, and extra explanation of implementation should be added.

I would consider changing the title to the manuscript to something more reflective of the end goals of the paper. While a review of past heterogeneity measures is vital to introducing new methods, I am confused as to why " in hydrologic studies" is used. The connotation seems to be that you are applying new methods to the results of past studies, which is not the case. Consider "New developments of heterogeneity measures for synthetic distributions of extreme hydrologic events."

Specific comments: Line 8 Page 1 - I found the first sentence "Regional frequency analysis is needed to..." to be misleading. While this statement is certainly true, I did not find this to be a major part of this study. This statement should be in the introduction as background instead.

---

## Author Comment (AC2) · 29 Aug 2016

Authors' reply to Anonymous Referee #3

The manuscript titled 'Heterogeneity measures in hydrological studies: review and new developments' presents a summary of the state current state of Regional Hydrologic Frequency Analysis (RHFA). Generally, I found the manuscript to be a very interesting and information dense product that I enjoyed reading. However, I think that there are missing components that limited my understanding of the implications of this study. This manuscript has a lot of ground to cover to get to it's results, and I encourage the authors to include key information and reorganize some of the sections as per my general comments below.

Reply: The authors thank the reviewer for the thorough revision of the manuscript as

[Figure]

[Figure]

well as for the constructive comments provided. The authors tried to address all the comments raised by the reviewer. Please, see the reply to general comments below.

1. I was not able to find the data source in which the study was applied. It seems as though the data might be synthetic and generated as a hypothetical, but that is not clear. Please include at lease a small section specifically about how the data was used (if measured from real data) or synthesized (if it was generated by the authors). Please include this data, or summary of data, either in the manuscript itself or as supplemental material.

Reply: The present study is based on synthetic data, as it is briefly described in the second paragraph of Sect. 2 "Assessment of a heterogeneity measure". In this regard, the authors agree with the reviewer about the need of including a specific section, e.g. Sect. 2.1 "Synthetic regions", to better underline this. Also, for clarity reasons, "simulation-based" will be added to an existing sentence in the Conclusions: "In the present paper, a simulation-based general framework is presented..." The authors understand the suggestion of the reviewer about including the synthetic data used or a summary of them. In this regard, a better description of the general way in which the regions are generated will be included in the aforementioned new Sect. 2.1. Nevertheless, the amount of data used in this study is very large and their characteristics change depending on the step of the methodology applied for their assessment. The authors believe that the current description of the specific values considered for generating the regions of study in each step of the methodology may be considered as enough for understanding the data used. The authors would like to note that many simulation studies present their results in a similar way (e.g. Viglione et al. 2007; Wright et al. 2015).

2. The Gini Index is a very popular index to determine economic equality as the authors mention, but there should be additional descriptions about why the Gini Index was applied in the way that it was. Many of the other methods have been used in the past and are presented as benchmarks. However, the Gini Index is fairly new in hydrologic studies, and extra explanation of implementation should be added.

Reply: The authors agree with the reviewer. The Gini Index has not been directly applied to hydrology. However, as mentioned in Sect. 3.3, it is connected with the well-known L-moments which do. Indeed, the Gini Index corrected for short samples corresponds to the sample L-CV. Then, the Gini Index applied on the at-site L-CV in a region provides a value of the dispersion of the at-site L-CV in such a region, and hence it can be seen as a measure of its heterogeneity. Sect. 3.3 will be rewritten for extending the description of the Gini Index and clarifying this point.

3. I would consider changing the title to the manuscript to something more reflective of the end goals of the paper. While a review of past heterogeneity measures is vital to introducing new methods, I am confused as to why " in hydrologic studies" is used. The connotation seems to be that you are applying new methods to the results of past studies, which is not the case. Consider "New developments of heterogeneity measures for synthetic distributions of extreme hydrologic events."

Reply: The authors thank the reviewer for pointing this out and agree that the use of "in hydrologic studies" may be confusing. They also thank the reviewer for the new title suggestion. In this regard, the authors prefer not to use "synthetic distributions" in the title to avoid misunderstandings. Note that in this study the heterogeneity measures are assessed by using simulated data, but they will be applied on real data in practice. The synthetic data simulate real hydrologic conditions and in no way restrict the scope of the analysis. The title of the paper will be changed from "Heterogeneity measures in hydrological studies: review and new developments" to "Heterogeneity measures in hydrological frequency analysis: review and new developments", based on the reviewer suggestion. The authors wish again to thank the reviewer for his important input concerning the title of the paper.

4. Specific comments: Line 8 Page 1 - I found the first sentence "Regional frequency analysis is needed to..." to be misleading. While this statement is certainly true, I did not find this to be a major part of this study. This statement should be in the introduction as background instead.

[Figure]

Reply: The authors agree with the reviewer. This sentence will be removed from the abstract and adapted to the Introduction.

References:

Viglione, A., Laio, F. and Claps, P.: A comparison of homogeneity tests for regional frequency analysis, Water Resour. Res., 43, 2007.

Wright, M.J., Houck, M.H. and Ferreira, C.M.: Discriminatory Power of Heterogeneity Statistics with Respect to Error of Precipitation Quantile Estimation, J. Hydrol. Eng., 04015011, 2015.

---

## Author Comment (AC3) · 29 Aug 2016

Authors' reply to Anonymous Referee #2

GENERAL COMMENTS

I really enjoyed reading this paper. I appreciate the efforts taken by the authors in contributing to the field of Regional Frequency analysis (RFA) by proposing a new index to assess heterogeneity of a region. There is clarity in describing the assumptions/drawbacks associated with the existing heterogeneity measures. Also, it is nicely stated why there is a need for new heterogeneity measures in RFA. The criteria that are defined to compare various heterogeneity measures with a new measure look adequate.

Reply: The authors thank the reviewer for the thorough revision as well as for the constructive comments provided.

SPECIFIC COMMENTS

1. In the introduction section, literature review regarding regional hydrological frequency analysis is not complete.

Reply: The literature review on regional hydrological frequency analysis included in the introduction of this study is not intended to be exhaustive. Its aim is to provide a general idea of the diversity of tools and methods existing for performing regional hydrological frequency analysis, with the aim of underlining the need for heterogeneity measures for the evaluation of the corresponding regional heterogeneity. In order to guide the reader in the search of additional references, "see Ali et al., 2012" will be changed to "see Ali et al., 2012 and references herein" (page 2 line 6). Also, the sentence "For further references on regional flood frequency analysis, please see Ouarda (2013), Salinas et al. (2013) and references herein" will be added to page 2 line 10.

2. Page 3, Line 32, "... focused on the delineation step". What do you mean by "..focussed on delineation step"? How Gini Index (proposed in this paper) account for the delineation step?

Reply: The authors apologise for the lack of clarity. The intention was to highlight that the use of a heterogeneity measure will allow direct comparison of the heterogeneity of regions delineated by different methods. Please, see reply to comment 6 for additional explanations about the use of the Gini Index for comparing the heterogeneity of regions delineated by different delineation methods. For clarity, "... and focused on the delineation step" will be replaced by the aforementioned explanation "Furthermore, the use of a heterogeneity measure will allow direct comparison of the heterogeneity of regions delineated by different methods".

3. Page 4, line 28-29; what is the reason behind considering a linear relationship for

L-CV and why gamma/2 was used?

Reply: This linear relation and the use of gamma/2 is a common and plausible way of simulating a varying at-site L-CV over a region. It has been used in other studies, such as Hosking and Wallis (1997) and Wright et al. (2015). In this regard, the following sentence will be added to page 4 line 29: "Note that this relation is commonly used in other studies (e.g. Hosking and Wallis, 1997; Wright et al., 2015) as a plausible way of simulating varying conditions over a region".

4. Page 10, line 25-26; Gini index considered in this study is a function L-CV . Can GI be expressed in terms of L-skewness coefficient on the similar line as it was expressed in terms of L-CV? If yes, then will there be any change in the overall results from this study? I am asking this question because L-skewness would be more uncertain for observed data (due to involvement of higher (third) moment) which can influence the heterogeneity measure H2 or V2 (e.g variation in (V2) H2 heterogeneity measure would be more as compared to (V1) H1 heterogeneity measure; even visible in Figures 1, 2 and 3). As the indices are compared based on hypothetical regions (using Monte Carlo simulation), the effect on L-skewness coefficient may not be noticeable. But in practice, there can be uncertainty associated with the estimation of L-skewness coefficient.

Reply: The GI has been expressed in terms of the L-CV due to the fact that the L-CV has been recognised as a good surrogate of the difference between at-site flood distributions. It also has a physical meaning, as it is related to the slope of the frequency curve. In theory, the GI could be expressed in terms of the L-skewness as well. However, as mentioned by the reviewer, the use of the L-skewness would imply high uncertainty when applying it to real data. Note that the length of observed data is usually short to obtain an appropriate estimation of the L-skewness in practice, making the use of a lower L-moment such as the L-CV preferable. Also note that when traditionally assessing the homogeneity of a region, the use of the H1 statistic (only based on L-CV) has been generally recognised as more powerful than others such as H2 (based on L-CV and on L-skewness), and that to our knowledge, statistics only based

on L-skewness are not usually considered.

5. What is the range GI index?

Reply: This can be found in Sect. 3.3, after Eq. (10): "Theoretically, GI ranges from zero to one. The former is obtained when all the $x_i$ values are equal, and the latter is given when all but one value equals zero (in an infinite population)".

6. Page 16, line 3-10 describes use of GI in assessing the delineation methods. The methodology discussed is not clear. Different delineation methods have different criteria to select the optimal delineation/regionalization solution (e.g. AIC or BIC, Davies-Bouldin index in the case of hard clustering algorithms, Xie-Beni in the case of fuzzy clustering algorithm). Are you trying to say that we can rank the delineation methods based on GI index value to arrive at best delineation method for regionalization which then can be used to perform regional frequency analysis? If yes, then I think this kind of approach is even possible with any other Index (e.g., H1, H2). Authors have mentioned on Page 3, line 32 that the proposed index (e.g., GI) can alleviate the drawback associated with delineation method as compared to the conventional heterogeneity indices (H1, H2 etc).

Reply: The authors apologise for the lack of clarity. The GI is intended to be used in assessing delineation methods by comparing the heterogeneity of the regions already identified as the "best regions" by each delineation method. For instance, each one of two different delineation methods (methodA and methodB) delineates two different sub-regions (sub-regionA1 and sub-regionA2; sub-regionB1 and sub-regionB2). The GI will then be applied on the at-site L-CV obtained from the flood data of each sub-region (GIA1, GIA2, GIB1, GIB2); and the GI for each delineation method will be obtained as the GI average of the corresponding sub-regions (GIA, GIB). Then, the best delineation method will be the one with the smallest GI (i.e., min(GIA,GIB)), as it obtains the less heterogeneous sub-regions. For clarity, this example will be properly incorporated into the Discussion section. Note that the GI is identified as the best measure for ranking

regional heterogeneity in this study, as it is the one with the best behaviour over the four steps of the assessment procedure.

Regarding the comment of the reviewer concerning the text in page 3 line 32, the authors also apologise for the lack of clarity. Such a drawback was not related to conventional heterogeneity indices, but to procedures that imply the need of performing additional steps in the regional analysis for comparing delineations methods. Please, see reply to specific comment 2 for further explanations.

7. In Figure 7, impact of heterogeneity due to addition of discordant sites to homogeneous regions is presented. A similar observation can be concluded from Figures 1 where the indices are assessed with increase in the heterogeneity percent (gamma value). In the case of Figure 1, mean of conventional heterogeneity measures (H1, H2, AD) tend to diverge with increase in the degree of heterogeneity while in Figure 7, Heterogeneity measures tend to converge (coming close) with increase in the number of discordant sites (i.e. increase in the degree of heterogeneity). Kindly clarify this point. However, in the case of GI, in both figures1 and 7, convergence is visible. Why is this happening?

Reply: It is important to highlight that, although both graphics have in common an increase in the regional heterogeneity, their characteristics and displayed information are different. In the case of Fig. 1, heterogeneous regions with increasing heterogeneity (from 0% to 100%) are generated by considering two different regional L-CVs and their heterogeneity values are compared. As a result, measures such as H1 and H2 "diverge" with increasing heterogeneity, as these measures are not originally thought to compare several (heterogeneous) regions. In the case of Fig. 7 the situation is different. A given homogeneous region is considered, and "homogenous" sites are exchanged by "discordant" sites. Hence, in this case, progressive changes in the heterogeneity of a given region are assessed (not between different regions). As a result, the heterogeneity of the region is more difficult to be ranked for all measures when there is an increase in the number of discordant sites and in the difference between L-

CVs (of homogeneous region and discordant sites). For clarity, the following sentence will be added to the Sect. 4.4, after the existing description of Fig. 7: "Note that unlike Fig. 1, where the heterogeneity value of two kinds of regions with the same degree of heterogeneity but different regional L-CV may be compared; in Fig. 7 progressive changes in the heterogeneity of a single homogeneous region are assessed."

8. The results obtained from the Monte Carlo simulation study are encouraging. Only thing missing from the paper is to perform analysis using observed data which would help to strengthen the conclusions drawn from the study. Analysis based on observed data would provide more clarity on the aforementioned concerns especially on Specific comment 4.

Reply: The authors agree with the reviewer on the need to extend the present study to observed data. However, the present manuscript is itself a very dense paper, and such application would imply the delineation of homogenous regions for a given case study by using different delineation methods, which may be considered as a whole new study. The authors would also like to highlight that many studies over the literature compared measures by using simulations without including a real case study (e.g. Genest et al. 2009). Furthermore, in a real case study, the performance of the measures cannot be evaluated because of the absence of the reference heterogeneity value (unlike in simulations). The authors believe that including a case study may not provide an added value to the paper but could have a negative impact in terms of clarity and readability, while making the manuscript even longer. In this regard, the following sentence will be added to the last paragraph of Conclusions: "In this study, the performance of heterogeneity measures is evaluated through a simulation-based procedure and recommendations are given. Further research may focus on their application to a variety of case studies in order to analyse practical aspects."

TECHNICAL CORRECTIONS

1. First line of Abstract reads "Regional frequency analysis is needed to estimate hydrological quantiles at ungauged sites or to improve estimates at sites with short record lengths, by transferring information from gauged sites." I think the regional transfer of information is possible with hydrologically similar sites only. Hence, ". . .hydrologically similar gauged sites" instead of ". . . gauged sites" looks more appropriate.

Reply: The suggestion of the reviewer will be taken into account when moving the aforementioned sentence from the abstract to the Introduction as suggested by Referee#3. The authors thank the reviewer for this comment.

References:

Genest, C., Rémillard, B. and Beaudoin, D.: Goodness-of-fit tests for copulas: A review and a power study, Insurance: Mathematics and economics, 44(2), 199-213, 2009.

Hosking, J.R.M. and Wallis, J.R.: Regional frequency analysis: an approach based on L-moments, Cambridge University Press, 240 pages, 1997.

Ouarda, T.B.M.J.: Hydrological Frequency Analysis, Regional. Encyclopedia of Environmetrics, John Wiley & Sons, Ltd, 2013.

Salinas, J., Laaha, G., Rogger, M., Parajka, J., Viglione, A., Sivapalan, M., et al.: Comparative assessment of predictions in ungauged basins–Part 2: Flood and low flow studies, Hydrology and Earth System Sciences, 17, 2637-2652, 2013.

Wright, M.J., Houck, M.H. and Ferreira, C.M.: Discriminatory Power of Heterogeneity Statistics with Respect to Error of Precipitation Quantile Estimation, J. Hydrol. Eng., 04015011, 2015.

---

## Author Response (AR1)

**Authors' reply to Editor and Referees "Heterogeneity measures in hydrological frequency analysis: review and new developments" by A. I. Requena et al.**

**A) Authors' reply to Editor Dr. Stacey Archfield**

Editor Decision: Reconsider after major revisions

COMMENTS TO THE AUTHOR:

1. This manuscript received three constructive reviews and, at this time, I instruct the authors to make their proposed changes to revise the manuscript. Given the feedback of the reviews and the required changes, the manuscript will be sent back out for review once the revisions are made. Final acceptance is not guaranteed.

**Reply**: The authors thank the editor for managing the present manuscript, as well as for her constructive comments. The revised version of the manuscript where the changes described in the present document are included has been submitted for its assessment.

2. All reviewers noted that a lack of application to observed data detracted from the manuscript. In my reading of the manuscript, the comments, and the responses, I share in their comment. I appreciate the need to perform simulated experiments to evaluate the various heterogeneity measures and recognize that simulation experiments have been published alone in previous papers. However, with regional frequency analysis now having well-established and widely accepted practices, assessing the overall effect of any proposed change in the overall practice would provide an even stronger case for use of a different method in assessing heterogeneity.

For this reason, I strongly encourage the authors to think about adding a section that applies the Gini Index to observed data. As the authors note in their reply, assessing heterogeneity is only one step in a multi-step process. Perhaps the authors' experience with regional flood frequency analysis has resulted in a readily available dataset for which the Gini Index could be applied and

contrasted with previously-computed heterogeneity metrics and, ultimately, regional flood frequency estimates to determine the overall effect the Gini Index has.

**Reply**: The authors understand the point of the editor and referees. A new Sect. 6 entitled "Illustrative application" has been included in the revised manuscript. In such a section, the GI and commonly used criteria are applied on a real case study for didactical purposes, and discussion about the obtained results is provided. Please, see Authors' reply to Specific Comment 8 of Referee#2 and new Sect. 6 in the revised manuscript.

**B) Authors' reply to Referee #1 C. Rojas-Serna**

It's a good idea to determine the homogeneous zones from the identification of the heterogeneity between both, nevertheless, the amount of data used in this work is poor. This has consequences in the discussion of results because is not clear which is the hydrological variable that determines the heterogeneity. My comment is focused in the data used in this work: There is not a section where data used to apply the methodology are presented. This limits the understanding of the heterogeneity in hydrology. It is unclear whether this is a watershed or region where the variable n is gauging stations. Or maybe n may be the number of meteorological stations. In that case the heterogeneity would be defined by climatic characteristics. I suggest include a section where the study area and the hydrological variable that is considered to define heterogeneity will be presented. So also describe the importance of data in the discussion of the results.

**Reply**: The authors thank the reviewer for the time spent in this revision as well as for her comments. The present study is based on synthetic regions that are simulated by a Monte Carlo procedure. The definition of these regions was presented in the second paragraph of Sect. 2 "Assessment of a heterogeneity measure" of the original manuscript. First, the authors agree with the reviewer that this can be more clearly highlighted as a new Sect. 2.1 "Synthetic regions", which has been included in the revised version of the manuscript. In addition, the following words marked between hyphens have also been added to the text: "The procedure is based on synthetic regions generated through Monte Carlo simulations from a representative – flood – parent distribution […]." (page 4 line 21 of the revised manuscript); "A region is defined by its number of – gauging – sites (N), at-site data length (n), […]" (page 4 line 22 of the revised manuscript).

Since the regions are simulated, the amount of available flood data is very large and the regions present a variety of situations. This can be seen for instance by looking at Figs. 1, 2 and 3. A large amount of simulated data is essential to perform the proposed assessment procedure through which the potential heterogeneity measures are evaluated according to different factors and criteria.

Finally, the authors think that it is important to highlight the purpose of this study. The heterogeneity measure identified by the four-step assessment procedure proposed in this study will be used for quantifying the degree of heterogeneity of a given region. This is a part of a whole regional frequency analysis in which a given region would have been previously defined by using climatic, hydrologic and/or physiographic descriptors, a given delineation method, etc (e.g. see new Sect. 6 "Illustrative application" in the revised manuscript). The heterogeneity measure will be then applied on the flood data of the given defined region to quantify its degree of regional heterogeneity, in an analogous way to when homogeneity tests are applied. Note that the heterogeneity of a given region in this study, as well as in the literature, is based on differences in any feature of the at-site frequency distribution among sites (e.g. Hosking and Wallis, 1997), where the L-variation coefficient may be considered as representative of them (e.g. Viglione, 2010). This is the reason of the use of simulated flood data in this study. This information can be found over the manuscript (Sect. 1, 2 and 5).

The use of simulated data in the assessment of new techniques in regional frequency analysis is a very well established approach and it has been used in a number of publications (for instance Hosking and Wallis, 1997; Seidou et al., 2006; Chebana and Ouarda, 2007). In fact, this is the only way to deal with issues related to data quality. These issues were rightfully raised by the reviewer. In this regard, the aforementioned explanation has been added at the end of new Sect. 2.1: "It is important to highlight that the use of simulated data in the assessment of new techniques in regional frequency analysis is a very well established approach and it has been used in a number of publications (e.g. Hosking and Wallis, 1997; Seidou et al., 2006; Chebana and Ouarda, 2007). Indeed, this is the only way to deal with issues related to data quality".

If observed data is used, we will be faced with issues related to the size of the record, sampling uncertainty, ignorance of parent distribution, ignorance of the true nature of the link between physiographical-meteorological variables and hydrological ones, etc. The authors agree that this element needed to be more clearly presented in the manuscript. Please, see the illustrative application on a real case study that has been added for didactical purposes as a new Sect. 6, where discussion is also shown. The authors wish to thank the reviewer for having pointed out this element.

References:

Chebana, F. and Ouarda, T.B.M.J.: Multivariate L-moment homogeneity test, Water Resources Research. 43, W08406, 2007.

Hosking, J.R.M and Wallis, J.R.: Regional frequency analysis: an approach based on L-moments, Cambdridge University Press, 240 pages, 1997.

Seidou, O., Ouarda, T.B.M.J., Barbet, M., Bruneau, P. and Bobée, B.: A parametric Bayesian combination of local and regional information in flood frequency analysis, Water Resources Research. 42, W11408, 2006.

Viglione, A.: Confidence intervals for the coefficient of L-variation in hydrological applications, Hydrol. Earth Syst. Sc., 14, 2229-2242, 2010.

**C) Authors' reply to Anonymous Referee #2**

GENERAL COMMENTS

I really enjoyed reading this paper. I appreciate the efforts taken by the authors in contributing to the field of Regional Frequency analysis (RFA) by proposing a new index to assess heterogeneity of a region. There is clarity in describing the assumptions/drawbacks associated with the existing heterogeneity measures. Also, it is nicely stated why there is a need for new heterogeneity measures in RFA. The criteria that are defined to compare various heterogeneity measures with a new measure look adequate.

**Reply**: The authors thank the reviewer for the thorough revision as well as for the constructive comments provided.

SPECIFIC COMMENTS

1. In the introduction section, literature review regarding regional hydrological frequency analysis is not complete.

**Reply**: The literature review on regional hydrological frequency analysis included in the introduction of this study is not intended to be exhaustive. Its aim is to provide a general idea of the diversity of tools and methods existing for performing regional hydrological frequency analysis, with the aim of underlining the need for heterogeneity measures for the evaluation of the corresponding regional heterogeneity. In order to guide the reader in the search of additional references, "see Ali et al., 2012" has been changed to "see Ali et al., 2012 and references herein" (page 2 line 7 of the revised manuscript). Also, the sentence "For further references on regional flood frequency analysis, please see Ouarda (2013), Salinas et al. (2013) and references herein" has been added to page 2 lines 11-12 of the revised manuscript.

2. Page 3, Line 32, "… focused on the delineation step". What do you mean by "..focussed on delineation step"? How Gini Index (proposed in this paper) account for the delineation step?

**Reply**: The authors apologise for the lack of clarity. The intention was to highlight that the use of a heterogeneity measure will allow direct comparison of the heterogeneity of regions delineated by different methods. Please, see Authors' Reply to Specific Comment 6 for additional explanations about the use of the Gini Index for comparing the heterogeneity of regions delineated by different delineation methods. For clarity, "[…] and focused on the delineation step" has been replaced by the aforementioned explanation "Furthermore, the use of a heterogeneity measure should allow direct comparison of the heterogeneity of regions delineated by different methods" (page 3 lines 33-34 of the revised manuscript).

3. Page 4, line 28-29; what is the reason behind considering a linear relationship for L-CV and why gamma/2 was used?

**Reply**: This linear relation and the use of gamma/2 is a common and plausible way of simulating a varying at-site L-CV over a region. It has been used in other studies, such as Hosking and Wallis (1997) and Wright et al. (2015). In this regard, the following sentence has been added to page 5 lines 2-4 of the revised manuscript: "Note that this relation is commonly used in other studies (e.g. Hosking and Wallis, 1997; Wright et al., 2015) as a plausible way of simulating varying conditions over a region".

4. Page 10, line 25-26; Gini index considered in this study is a function L-CV . Can GI be expressed in terms of L-skewness coefficient on the similar line as it was expressed in terms of L-CV? If yes, then will there be any change in the overall results from this study? I am asking this question because L-skewness would be more uncertain for observed data (due to involvement of higher (third) moment) which can influence the heterogeneity measure H2 or V2 (e.g variation in (V2) H2 heterogeneity measure would be more as compared to (V1) H1 heterogeneity measure; even visible in Figures 1, 2 and 3). As the indices are compared based on hypothetical regions (using Monte Carlo simulation), the effect on L-skewness coefficient may not be noticeable. But in practice, there can be uncertainty associated with the estimation of L-skewness coefficient.

**Reply**: The GI has been expressed in terms of the L-CV due to the fact that the L-CV has been recognised as a good surrogate of the difference between at-site flood distributions. It also has a

physical meaning, as it is related to the slope of the frequency curve. In theory, the GI could be expressed in terms of the L-skewness as well. However, as mentioned by the reviewer, the use of the L-skewness would imply high uncertainty when applying it to real data. Note that the length of observed data is usually short to obtain an appropriate estimation of the L-skewness in practice, making the use of a lower L-moment such as the L-CV preferable. Also note that when traditionally assessing the homogeneity of a region, the use of the H1 statistic (only based on L-CV) has been generally recognised as more powerful than others such as H2 (based on L-CV and on L-skewness), and that to our knowledge, statistics only based on L-skewness are not usually considered.

5. What is the range GI index?

**Reply**: This can be found in Sect. 3.3, after Eq. (10): "Theoretically, GI ranges from zero to one. The former is obtained when all the $x_i$ values are equal, and the latter is given when all but one value equals zero (in an infinite population)".

6. Page 16, line 3-10 describes use of GI in assessing the delineation methods. The methodology discussed is not clear. Different delineation methods have different criteria to select the optimal delineation/regionalization solution (e.g. AIC or BIC, Davies- Bouldin index in the case of hard clustering algorithms, Xie-Beni in the case of fuzzy clustering algorithm). Are you trying to say that we can rank the delineation methods based on GI index value to arrive at best delineation method for regionalization which then can be used to perform regional frequency analysis? If yes, then I think this kind of approach is even possible with any other Index (e.g., H1, H2). Authors have mentioned on Page 3, line 32 that the proposed index (e.g., GI) can alleviate the drawback associated with delineation method as compared to the conventional heterogeneity indices (H1, H2 etc).

**Reply**: The authors apologise for the lack of clarity. The GI is intended to be used in assessing delineation methods by comparing the heterogeneity of the regions already identified as the "best regions" by each delineation method. For instance, each one of two different delineation methods (methodA and methodB) delineates two different sub-regions (sub-regionA1 and sub-regionA2; sub-regionB1 and sub-regionB2). The GI will then be applied on the at-site L-CV obtained from

the flood data of each sub-region (GIA1, GIA2, GIB1, GIB2); and the GI for each delineation method will be obtained as the GI average of the corresponding sub-regions (GIA, GIB). Then, the best delineation method will be the one with the smallest GI (i.e., min(GIA,GIB)), as it obtains the less heterogeneous sub-regions. This is clearly shown in new Sect. 6 "Illustrative application". For guiding the reader in this regard, the sentence "(see Sect. 6 for an illustrative application)" has been included in Discussion (page 16 line 26 of the revised manuscript). Note that the GI is identified as the best measure for ranking regional heterogeneity in this study, as it is the one with the best behaviour over the four steps of the assessment procedure.

Regarding the comment of the reviewer concerning the text in page 3 line 32 of the original manuscript, the authors also apologise for the lack of clarity. Such a drawback was not related to conventional heterogeneity indices, but to procedures that imply the need of performing additional steps in the regional analysis for comparing delineations methods. Please, see Author' Reply to Specific Comment 2 for further explanations.

7.  In Figure 7, impact of heterogeneity due to addition of discordant sites to homogeneous regions is presented. A similar observation can be concluded from Figures 1 where the indices are assessed with increase in the heterogeneity percent (gamma value). In the case of Figure 1, mean of conventional heterogeneity measures (H1, H2, AD) tend to diverge with increase in the degree of heterogeneity while in Figure 7, Heterogeneity measures tend to converge (coming close) with increase in the number of discordant sites (i.e. increase in the degree of heterogeneity). Kindly clarify this point. However, in the case of GI, in both figures1 and 7, convergence is visible. Why is this happening?

**Reply**: It is important to highlight that, although both graphics have in common an increase in the regional heterogeneity, their characteristics and displayed information are different. In the case of Fig. 1, heterogeneous regions with increasing heterogeneity (from 0% to 100%) are generated by considering two different regional L-CVs and their heterogeneity values are compared. As a result, measures such as H1 and H2 "diverge" with increasing heterogeneity, as these measures are not originally thought to compare several (heterogeneous) regions. In the case of Fig. 7 the situation is different. A given homogeneous region is considered, and "homogenous" sites are exchanged by "discordant" sites. Hence, in this case, progressive changes in the heterogeneity of a given region are assessed (not between different regions). As a

result, the heterogeneity of the region is more difficult to be ranked for all measures when there is an increase in the number of discordant sites and in the difference between L-CVs (of homogeneous region and discordant sites). For clarity, the following sentence has been added after the existing description of Fig. 7 to the first paragraph in Sect. 4.4: "Note that unlike Fig. 1, where the heterogeneity value of two kinds of regions with the same degree of heterogeneity but different regional L-CV may be compared; in Fig. 7 progressive changes in the heterogeneity of a single homogeneous region are assessed."

8.   The results obtained from the Monte Carlo simulation study are encouraging. Only thing missing from the paper is to perform analysis using observed data which would help to strengthen the conclusions drawn from the study. Analysis based on observed data would provide more clarity on the aforementioned concerns especially on Specific comment 4.

**Reply**: The authors agree with the reviewer on the need to extend the present study to observed data. However, the authors believe that the present manuscript is itself a very dense paper, and such application would imply the delineation of homogenous regions for a given case study by using different delineation methods, which may be considered as a whole new study. The authors would also like to highlight that many studies over the literature compared measures by using simulations without including a real case study (e.g. Genest et al. 2009). Furthermore, in a real case study, the performance of the measures cannot be evaluated because of the absence of the reference heterogeneity value (unlike in simulations).

Nevertheless, for didactical purposes and encouraged by the editor, the authors have decided to include an "illustrative application" as new Sect. 6. In such a section, the GI and commonly used criteria for assessing the performance of delineation methods are applied on a real case study. As illustration, simple clustering settings are used for delineating the regions and discussion about results is provided. Please, see new Sect. 6 in the revised manuscript.

In this regard, the following sentences have also been added to the revised manuscript. Sentence added to the last paragraph of Conclusions: "In this study, an illustrative application is also included for didactical purposes. The subjectivity related to commonly used criteria in assessing the performance of different delineation methods is underlined through it. In this regard, further research may also focus on the application of heterogeneity measures to a variety of case studies

in order to analyse practical aspects". Sentence added at the end of the abstract: "An illustrative application is also presented for didactical purposes, through which the subjectivity of commonly used criteria in assessing the performance of different delineation methods is underlined". Regarding Specific Comment 4, please see the corresponding Authors' Reply to that point.

TECHNICAL CORRECTIONS

1. First line of Abstract reads "Regional frequency analysis is needed to estimate hydrological quantiles at ungauged sites or to improve estimates at sites with short record lengths, by transferring information from gauged sites." I think the regional transfer of information is possible with hydrologically similar sites only. Hence, "…hydrologically similar gauged sites" instead of "… gauged sites" looks more appropriate.

**Reply**: The suggestion of the reviewer has been taken into account when moving the aforementioned sentence from the abstract to the Introduction as suggested by Referee#3: "[…]. This is done by transferring information from hydrologically similar gauged sites" (page 1 line 27 of the revised manuscript). The authors thank the reviewer for this comment.

**Reply**: The authors agree with the reviewer. The sentence has been removed from the abstract; and for clarity, the words marked between hyphens have been added to the sentence "Some regional procedures - to estimate hydrological quantiles at ungauged sites - , such as the index-flood method, […]" (page 1 line 8 of the revised manuscript). Moreover, the sentence removed from the abstract has been adapted to the introduction as follows "[…]. 
[revised manuscript text omitted]

---

## Author Response (AR2)

**Authors' reply to Editor and Referees "Heterogeneity measures in hydrological frequency analysis: review and new developments" by A. I. Requena et al.**

**A) Authors' reply to Editor Dr. Stacey Archfield**

Editor Decision: Publish subject to minor revisions (further review by Editor)

COMMENTS TO THE AUTHOR:

All reviewers acknowledge the revisions have improved the manuscript and the authors have addressed many of the technical comments raised in the first set of reviews; however, two of the reviewers felt additional revision is needed to further clarify and strengthen the presentation of the research.

At this point, I would ask the authors to carefully respond and revise the manuscript based on this set of reviews. After that is complete, I will make a final decision on the manuscript.

**Reply**: The authors thank the editor for managing the present manuscript, as well as for her thorough and constructive comments over the revision process. All comments provided by the reviewers have been addressed below, and corresponding changes have been included in the revised manuscript. Please, see Authors' reply to Reviewers for details.

**B) Authors' reply to Anonymous Referee #2: Report #1**

I recommend acceptance of the updated manuscript as it is.

**Reply**: The authors thank the reviewer for the time spent in the revision process of this manuscript and the constructive comments provided over this process.

**C) Authors' reply to New Referee #3 Keith Sawicz: Report #3**

The manuscript titled "Heterogeneity measures in hydrological frequency analysis: review and new developments." The study explores measures of heterogeneity on synthesized data, and I believe that it's contribution is value to the hydrologic sciences. The authors have improved the clarity of data synthesis methods from its last iteration, however I believe that additional revision is necessary before publication. I have included specific comments as examples of how to revise the manuscript for clarity and simplicity.

**Reply**: The authors thank the reviewer for the thorough revision and for the useful comments provided for the improvement of the manuscript.

1.  Page 1 Line 21: "…used criteria in assessing…" should be changed to "used criteria to assess"

**Reply**: The change has been done in the revised manuscript.

2.  Page 1 Line 28: "homogenous regions" has not been defined before this point. Care needs to be taken to use this nomenclature and I think that it is being used inappropriately in this case. Consider either replacing homogeneous with terminology used in the previous sentence ("hydrologically similar gauged sites"). Page 2 Line 3 has a definition of regional homogeneity that defines what the term means. Please define this term before using terminology like "homogeneous regions."

**Reply**: The authors thank the reviewer for pointing this out. The first paragraph of the Introduction has been rewritten following his suggestions. Please, see the revised manuscript.

3. Page 2 Line 9: Change "Known statistical tools" to "Traditional statistical tools."

**Reply**: The change has been done in the revised manuscript (P2, L10).

4. Page 2 Line 21-22: I do not believe that there is a definition of what you mean by "the quantile estimate." Please define prior to this statement.

**Reply**: The definition has been included in P1, L30 – P2, L1 of the revised manuscript: "[…] to estimate the magnitude of extreme events related to a given probability (or return period) at a target site, which are called quantiles." Note that "quantile estimate" has been changed to "quantile estimation" (P2, L23-24 in the revised manuscript).

5. Page 3 Line 11: The sentence containing "As a consequence of the non-availability…from the quantile estimate step." is overly complex. Consider simplifying the sentence structure to convey a clear point.

**Reply**: The sentence has been rewritten as follows (P3, L13-15 in the revised manuscript): "These studies usually consider measures based either on $H$ or on errors from the quantile estimation step. The reason is the non-availability of a well-justified heterogeneity measure for comparison purposes (approach (ii))."

6. Page 3 Line 26: I am not sure which approach you are referring to here when you say "…the latter approach…" The structure just before this sentence does not make it clear.

**Reply**: The authors apologise for the lack of clarity. "The latter approach" has been changed to "comparing quantile errors" (P3, L28 in the revised manuscript).

7. Page 5 Line 9: The sentence containing "the value of tau3 is (usually) omitted, as tau3 is considered to have the same value as tau." seems to either be a finding from this study or an

observation from other studies. If it is a finding of this study, this information is appropriate in the results, not the methods. If it is an observation from previous studies, a citation is needed.

**Reply**: The authors thank the reviewer for pointing out this blunder. The reference "(e.g. Hosking and Wallis, 1997)" has been included in this regard (P5, L14-15 in the revised manuscript).

8.  Page 5 Line 14: I agree that with the sentence containing "…the use of simulated data in the assessment of new techniques in regional frequency analysis is a very well established approach…" in that simulated data can be used to generally test new techniques because you can generate data with simple assumptions and not need to address uncertainty or bias in your data. However, while it is clear that data was simulated to test the new approach, it is not clear to me what kind of data was simulated. Was streamflow time series generated? Were synthetic flood frequency curves generated and sampled? You do not need to necessarily include the data that you generated, but it is important to specify the steps that you used to generate said data. The assumption is that when these steps are followed, you will arrive at the same conclusions of the paper even if the exact same data was not synthesized.

**Reply**: The authors apologize for the lack of clarity concerning this element. The information was in fact presented in the first paragraph of Sect. 2.1 "Synthetic regions", which mentioned that "The procedure is based on synthetic regions generated through Monte Carlo simulations from a representative flood parent distribution […]". For clarity, this sentence has been rewritten as follows in the revised manuscript (P4, L24-26): "The procedure is based on synthetic regions with flood data samples generated through Monte Carlo simulations from a representative flood parent probability distribution […]". Such a parent distribution is built under the general characteristics defined in Sect. 2.1, and under the particular characteristic described when performing each step of the methodology.

9.  Page 5 Line 15: "Indeed, this is the only way to deal with issues related to data quality". I don't agree with this statement. In order to simulate data, you are making assumptions about hydrologic systems by the GEV distribution. Like is mentioned in the manuscript, data

simulation is commonly used to test new approaches, but this is usually to address sampling bias and remove features of real data such as data availability/length and measurement uncertainty.

**Reply**: The authors agree with the reviewer. To avoid confusion, this statement is removed in the revised version of the manuscript.

**10.** Page 8 Line 9: "Heterogeneity measures". The previous section is titled 'assessment of heterogeneity measures,' but this section mentions the measures. Logically, I think you should introduce what you are going to compare before you talk about how you will compare them, unless there is a good reason otherwise.

**Reply**: The authors thank the reviewer and agree with him concerning this comment. However, the authors structured the manuscript in this manner with a specific objective: The authors believe that locating the section "assessment of heterogeneity measures" before "heterogeneity measures" helps to focus on the properties that a heterogeneity measure should have in the hydrological regional context, and also focus on the framework needed for their evaluation. Indeed, it is important to present the properties that heterogeneity measures should have, before proceeding with the development of these measures. Note that the other way around, placing first the section "heterogeneity measures", in which different kinds of measures from different approaches are defined and developed, may deviate the attention of the reader from the main focus of the paper.

**11.** Page 12 Line 11: In the sentence that says "all considered measures show a good behavior regarding the heterogeneity rate y, as their values increase with y.", good behavior is not a good way of describing this. These measures seem to be positively correlated with an increasing y, which would mean that all of the measures can indicate heterogeneity. However, the sentence as it is written is imprecise and vague. This is an example of sentence structure that should be simplified to better convey what you precisely mean and not rely on the reader to interpret what you mean.

**Reply**: The authors thank the reviewer for this comment. The sentence has been modified as follows (P12, L10-12 in the revised manuscript) "[…] all considered measures seem to be positively correlated with an increasing heterogeneity rate $\gamma$. This means that their behaviour is

appropriate as they may indicate heterogeneity". According to the suggestion of the reviewer, other sentences have been revised for their improvement. Some of the adjectives have been kept to help the reader understand the implications of a given dependence or effect. Please, see changes in Sect. 4.1 in the revised manuscript.

**12.** Table 1: I have not found where the definition of what is considered good, acceptable, or bad. Was there a numerical threshold determined for each of these? Are they purely qualitative? This table is comparing different measures of heterogeneity, so showing the quantitative thresholds that you used is useful to the reader.

**Reply**: The classification "good, acceptable or bad" is linked to the first step of the assessment procedure called "sensitivity analysis". This step may be considered as a preliminary step for identifying which measures may be appropriate as heterogeneity measures in the regional hydrological context. The definition of "good, acceptable or bad" is then purely qualitative. A more exhaustive analysis with objective criteria is later performed through steps two to four.

The following sentences have been modified for additional clarity:

- P4, L18-19: "The first step is applied to all the studied heterogeneity measures […] and may be considered as preliminary".

- P13, L21-22: "As a result of the aforementioned qualitative sensitivity analysis results (see Table 1 for a summary), $V'$, PCI and GI are considered as potentially suitable heterogeneity measures."

**13.** Figures (General): Generally, I find the figures to be labeled well and clear to understand. There is a legend in the figures that identify the difference between gray and black lines. One addition to this legend should be a mention of gray and black colors with respect to y values.

**Reply**: The legend of the figures includes the information needed for their interpretation.

**D) Authors' reply to New Referee #4 William Farmer: Report #2**

The authors have provided a scientifically-sound and well-written manuscript presenting, through controlled, scientific experiments using synthetically generated data, a novel technique for assessing the heterogeneity of a region. The authors have sufficiently responded to all previous reviewer comments and the body of the manuscript has improved. Most of my concerns are minor, but a will address my most important concerns first.

**Reply**: The authors thank the reviewer for the thorough revision and for all the useful comments provided for the improvement of the manuscript.

1. The editor and the reviewers called for a section using "real data" or observational data. I think this has been added, but I actually think, in its present form, it detracts from the manuscript. The synthetic experiments provide a more controlled justification of the novel method. When observational data is presented we are confronted with the problem that the true underlying condition is completely unknown. Therefore, exercises based on observational data must be cleverly defined so as to best approximate an unknown truth; this is an incredibly challenging task. For example, no one can definitively state that one of the three observational approaches to regionalization presented here is inherently better than any other. One can make a strong argument, and the authors have attempted to ensure that some of the approaches seem unlikely, but the truth is not known. Therefore, the result cannot be taken as inconclusive evidence. In this way, the introduction of observational data detracts from the neat, controlled experiments using synthetic data.

**Reply**: The assessment procedure based on simulations is the tool presented in this study to identify which heterogeneity measure may be considered as the best one. As mentioned by the reviewer, previous reviewers and the editor strongly suggested the inclusion of a section based on real data as illustration. Due to the impossibility of knowing the degree of heterogeneity of a region based on real observations, the authors decided to present such an analysis as an "Illustrative application". The aim of that section is then to provide an example of how different measures are used in practice and their drawbacks, as well as to guide the reader in the use of the heterogeneity measure found to be the best according to the proposed assessment procedure. This

point was underlined in Sect. 6 (P17, L9-14 in the revised manuscript): "Note that due to the data used in this application are observed instead of simulated, the real degree of heterogeneity of the regions, as well as the real parent distribution of the data are unknown. Thus, it is not possible to truly compare the performance of the different heterogeneity measures. In this regard, it is important to remark that the purpose of this illustrative application is then to show that commonly used criteria for identifying the best method for delineating regions may be subjective, as well as to guide practitioners in the use of heterogeneity measures."

Nevertheless, the authors agree with the reviewer that the premise indicating one of the delineation methods to be better than the others should be relaxed, as this is just an assumption. The text in Sect. 6 "Illustrative application" has been modified accordingly. Please, see related changes in the revised manuscript.

**2.** I am not advocating for the rejection of observational data. Instead, I propose using this data or another constructed example to discuss the limitations of this approach in practice. The authors suggest that the optimal metric would identify approach A as optimal. What factors might affect this discovery in practice? Beyond the excellent sensitivity analysis, are there external uncertainties that may affect your, by analogy, Type I and Type II errors?

**Reply**: The purpose of the illustrative application has been better highlighted by relaxing the statement indicating Clustering A as the best setting. This has been motivated by the previous comment of the reviewer; as such a premise is indeed just an assumption. In this regard, the following sentence has been included at the end of Sect. 6: "It is worth mentioning that Clustering A could be ideally assumed to be the best setting for forming sub-regions, as it is based on relevant descriptors for flood frequency analysis. However, this would just be an assumption that cannot be verified due to the use of observed data". For more details, please see changes in Sect. 6 of the revised manuscript. Regarding Type I and Type II errors, please see Authors' reply to Comment 12 below.

**3.** In this same vein, I'd like to see the authors expand Table 4. While the GI metric identifies A when a simple average is used, the more-appropriate weighted average identifies approach B as optimal. Furthermore, the weighted-averaging of GI suggests that A, B nor C are

less heterogeneous than using the entire region. Only the RRMSE of the 100-year event identifies approach A consistently. What causes this difference? Is it meaningful? (NOTE: When I computed weighted averages for GI I got 0.093, 0.092, and 0.093 for A, B, and C respectively. For RRMSE of the 100-year event, I got 16.20, 18.00, and 17.20; for the 10-year event I got 5.89, 5.83, and 5.61.)

**Reply**: The authors thank the reviewer for the comment. Although a weighted average may seem more appropriate, it is important to analyse why a value should be weighted. H is a measure that is affected by the number of sites of a region, which may be the reason of sometimes attempting to avoid this issue by weighing it. Nevertheless, the GI is a measure not affected by the number of sites, and then a simple average may be considered as suitable to understand its behaviour. While on the contrary, computing a weighted average may distort results. With the aim of avoiding any misunderstandings, the weighted average of $H$ shown in Table 4 was removed in the revised version of the manuscript. Note that such results were not relevant for the discussion in Sect. 6.

**4.** Let me close my main comments by saying that the remainder of the manuscript is very well written and the scientific procedure is well devised and executed. The findings are both novel and interesting, meriting publication. In coordination with the editor's opinion, I would advise revising the narrative of the observational example.

**Reply**: The authors thank the reviewer for all the comments. Section 6 "Illustrative application" has been modified according to the suggestions given by the reviewer in this revision. Please, see changes in Sect. 6 in the revised manuscript.

(ADDITIONAL COMMENTS)

The following a several more minor comments that I feel may be worthy of consideration towards improving the manuscript. They are presented in loose sequential order.

**Reply**: The authors thank the reviewer for the thorough revision.

**5.** P1, L27: Is hydrologic transfer from similar sites the only approach?

**Reply**: The sentence has been rewritten as "This is usually done by transferring information from hydrologically similar gauged sites".

**6.** P3, L7: What were the results of Wright et al. (2015)? How are they useful to this discussion?

**Reply**: The reference Wright et al. (2015) is included here (P3, L9 in the revised manuscript) to highlight that the approach related to "the use of heterogeneity measures as a proxy of quantile error" (P3, L3 in the revised manuscript) is already studied in the literature, which is mentioned in P2, L27-28 in the revised manuscript: "This approach has already been studied, being closely related to the homogeneity test notion (e.g. Hosking and Wallis, 1997; Wright et al., 2014), which is further explained below".

Relevant results of Wright et al. (2014 and 2015) for the present study are later shown when describing the heterogeneity measures considered (Sect. 3.1):

- P9, L6-8: "[…] $V_2$ and $H_2$, are also included in this study. Their inclusion is motivated by recent results regarding the usefulness of $H_2$ for testing homogeneity when considering different thresholds from those of $H$ (Wright et al., 2014)".

- P9, L17 – P10, L2: Regarding the $AD$ statistic, "Wright et al. (2015) evaluated its performance as a heterogeneity measure regarding its ability to be a surrogate of the quantile error, yet obtaining a weak performance partially attributed to a possible influence of the procedure used for estimating errors".

**7.** P4, L29: This statement is unclear to me. While L-CV and L-skew may be correlated, the heterogeneity rate appears to be more akin to variability, being a range divided by an average. Is there evidence to support similar variabilities? Can you clarify this statement? At first I thought you meant that L-CV equaled L-skew, but this is certainly not correct. I mention my misunderstanding only as an example the need for clarity.

**Reply**: The values of the L-CV and L-skewness, as well as other factors considered in the study, are selected based on the literature and with the aim of providing a general view of the behaviour

of the studied measures in different circumstances (see Sect. 2.1). This selection is done due to the impossibility of performing an exhaustive analysis considering all possible values for each factor. Similar variabilities for the at-site L-CV and L-skewness are considered based on the literature, and based on the fact that sites with a large L-CV often have a large L-skewness (see e.g. Hosking and Wallis, 1997, page 68 and Table 4.1). Regarding possible values of L-CV and L-skewness, please see Figure 1 in Viglione et al. (2007).

To improve clarity, the original statement has been modified as follows (P5, L2-4 in the revised manuscript): "Since in practice large values of the L-skewness coefficient ($\tau_3$) are related to large values of the L-CV $\tau$, and based on studies in the literature (e.g. see Hosking and Wallis, 1997, page 68 and Table 4.1; Viglione et al. 2007, Figure 1), the same heterogeneity rate of $\tau$ is considered for $\tau_3$."

**8.** P5, L1: In this paragraph, it might be clearer to use display equations rather than in-line equations.

**Reply**: The authors thank the reviewer for the comment. However, in line with standard formatting practices for equations, it may be preferable to keep the two equations in-line, as they are not referenced later in the text, and many displayed equations are already included in the manuscript.

**9.** P6, L1: Would there be value to considering more heterogeneity rates?

**Reply**: Other values were considered in preliminary analyses, but only results related to $\gamma = 0\%$ and 50% are shown due to space limitations, and because they may be considered as representative of the behaviour of the measures. Recall that in Sect. 2.1 (P5, L16-18 in the revised manuscript), it is indicated "[…] values of the factors, as well as their varying values used below, are selected according to the literature and with the aim of providing a general view of the behaviour of the measures without excessively complicating the simulation study."

**10.** P11, E11: Could the average L-CV be replaced by the regional L-CV? What would be the effect?

**Reply**: In the present study, synthetic flood data are built by considering the same data length for each site. Hence no effect would be observed in this case.

**11.** P13, L2: It may be useful to consider revising the presentation of results to appear more objective. For example, "favorable" and "bad" behavior seems to approach a subjective presentation. I often struggle with this in my own writing, so I only pass this comment along in case you find it useful.

**Reply**: The authors thank the reviewer for this honest comment. The presentation of results was revised and changes were made. Some of the adjectives were kept to help the reader understand the implications of a given dependence or effect. Please, see changes in Sect. 4.1 of the revised version of the manuscript.

**12.** P13, L24: This error rate seems to be an analog to Type I and Type II errors in hypothesis testing. Would that be a useful way to think about this approach? Does one metric provide a smaller error rate in either case?

**Reply**: The authors thank the reviewer for this interesting point of view. The success rate is an objective criterion used for assessing the ability of a given heterogeneity measure to identify the most heterogeneous region. Its assessment is based on simulations where the heterogeneity rate of region A, $\gamma_A$, is always smaller that the heterogeneity rate of region, $\gamma_B$. If we think of the success rate as in hypothesis testing, we would realise that the way in which its assessment is done does not allow e.g. evaluating a type II error. Indeed, type II error consists in "accepting $H_0$ (i.e. $H_0$: $\gamma_A < \gamma_B$) when it is false" and would be "zero". If we really want to establish a connection between hypothesis testing and success rate, it could be considered that the success rate is related to the correct inference of "accepting $H_0$ when $H_0$ is true", which is $1 -$ Type I error. To avoid misunderstandings and confusion, the authors do not recommend establishing this link in the present work.

[revised manuscript text omitted]